# STBLLM: BREAKING THE 1-BIT BARRIER WITH STRUCTURED BINARY LLMS

**Peijie Dong**[1,†], **Lujun Li**[2,†], **Yuedong Zhong**[3], **Dayou Du**[1], **Ruibo Fan**[1], **Yuhan Chen**[1],
**Zhenheng Tang**[2], **Qiang Wang**[4,*], **Wei Xue**[2], **Yike Guo**[2,*], **Xiaowen Chu**[1,2*]
[1] HKUST(GZ)  [2] HKUST  [3] SYSU  [4] HIT(SZ)
{pdong212, ddu487, rfan404, ychen906}@connect.hkust-gz.edu.cn,
lilujunai@gmail.com, zhongyd6@mail2.sysu.edu.cn,
qiang.wang@hit.edu.cn, {zhtang.ml, weixue, yikeguo}@ust.hk
xwchu@hkust-gz.edu.cn

## ABSTRACT

In this paper, we present the first structural binarization method for LLM compression to less than 1-bit precision. Although LLMs have achieved remarkable performance, their memory-bound nature during the inference stage hinders the adoption of resource-constrained devices. Reducing weights to 1-bit precision through binarization substantially enhances computational efficiency. We observe that randomly flipping some weights in binarized LLMs does not significantly degrade the model's performance, suggesting the potential for further compression. To exploit this, our STBLLM employs an N:M sparsity technique to achieve structural binarization of the weights. Specifically, we introduce a novel Standardized Importance (SI) metric, which considers weight magnitude and input feature norm to more accurately assess weight significance. Then, we propose a layerwise approach, allowing different layers of the LLM to be sparsified with varying N:M ratios, thereby balancing compression and accuracy. Furthermore, we implement a fine-grained grouping strategy for less important weights, applying distinct quantization schemes to sparse, intermediate, and dense regions. Finally, we design a specialized CUDA kernel to support structural binarization. We conduct extensive experiments on LLaMA, OPT, and Mistral family. STBLLM achieves a perplexity of 11.07 at 0.55 bits per weight, outperforming the BiLLM by 3×. The results demonstrate that our approach performs better than other compressed binarization LLM methods while significantly reducing memory requirements. Code is released at https://github.com/pprp/STBLLM.

## 1 INTRODUCTION

The advent of large language models (LLMs), such as (Zhang et al., 2022a; Touvron et al., 2023a; Brown et al., 2020), has revolutionized the field of natural language processing (NLP) (Wei et al., 2022b). These powerful models exhibit remarkable performance, surpassing human capabilities in various domains (Wei et al., 2022a; Bubeck et al., 2023). However, the immense scale and complexity of LLMs present significant challenges in terms of memory requirements, hindering their widespread deployment, especially in resource-constrained environments. To address this issue, model compression techniques, such as quantization (Frantar et al., 2023; Lin et al., 2024a; Dong et al., 2023b), pruning (Meng et al., 2020), distillation (Li, 2022; Li & Jin, 2022; Xiaolong et al., 2023), and low-rank decomposition (Ashkboos et al., 2024), have gained increasing attention in reducing the computational footprint while preserving their performance. One promising approach is network binarization, the most aggressive quantization method. Binarization quantizes original floating-point weights with binary values ($-1$ or $+1$), significantly reduces memory storage.

Pioneering binarization methods (Rastegari et al., 2016; Liu et al., 2018) present customized binary structures and training paradigms for binarized neural networks (BNNs) in vision tasks. Building

---

[*]Corresponding authors. [†] Equal contribution.

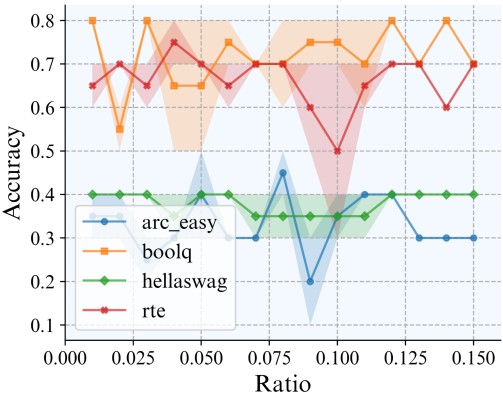

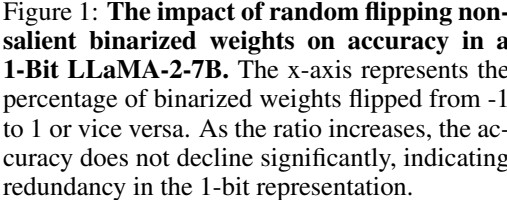

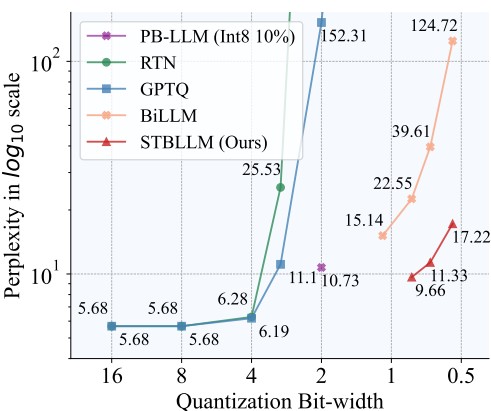

Figure 1: **The impact of random flipping non-salient binarized weights on accuracy in a 1-Bit LLaMA-2-7B.** The x-axis represents the percentage of binarized weights flipped from -1 to 1 or vice versa. As the ratio increases, the accuracy does not decline significantly, indicating redundancy in the 1-bit representation.

Figure 2: **The perplexity of LLaMA-1-13B on the Wikitext2 under different bit-widths.** RTN and GPTQ (Frantar et al., 2023) show a drastic performance drop at ultra-low bit-widths. Our STBLLM maintains low perplexity even at 0.55 bits compared with BiLLM.

upon these foundational approaches, subsequent methods (Wang et al., 2020; Liu et al., 2022; Munagala et al., 2020) have advanced the field by integrating sparse kernel techniques (Wang et al., 2023b; 2021b) and pruning methodologies (Wang et al., 2021a; Li & Ren, 2020; Munagala et al., 2020). For LLMs, inspired by the success of 4-bit and 8-bit quantization methods, some studies (Huang et al., 2024; Xu et al., 2024; Shang et al., 2024) continue to explore ultra-low-bit or even 1-bit precision. For example, the post-training method PB-LLM (Shang et al., 2024) partially binarizes LLMs with an optimal scaling factor strategy, preserving a small subset of the higher bit-precision weights. BiLLM (Huang et al., 2024) proposes a residual approximation strategy to improve 1-bit LLMs. While these methods represent the most aggressive quantization approaches, it is crucial to consider that popular floating-point LLMs already contain model sizes ranging from 7 billion to 140 billion parameters. As a result, 1-bit LLMs still need to be further accelerated and optimized for many resource-constrained devices and real-time scenarios. This naturally raises a key question: ***Is there any compression method with less than 1-bit weight representation that can further push the quantization of LLMs?***

For this question, there are two key observations: ① **Not all weights contribute equally to the performance of 1-bit LLMs.** As shown in Figure 1, performing random weight flipping for non-salient weights results in only a minimal performance drop (For more details, refer to Appendix B). This finding indicates that even in highly quantized 1-bit LLMs, a subset of redundant weights exists that can be compressed without impacting the performance. It suggests the potential for further compression by selectively encoding the most significant weights while discarding or compressing the less important ones. ② **Structured sparsity techniques,** such as N:M sparsity methods (Hubara et al., 2021; Zhang et al., 2022b; Zhou et al., 2021), leverage the inherent structure and patterns in the weight distribution, allowing for more efficient compression. These N:M sparsity methods have good hardware-accelerated support in recent LLM pruning models (Frantar & Alistarh, 2023; Sun et al., 2024; Dong et al., 2024), enabling efficient deployment on NVIDIA Ampere architecture (Nvidia, 2020). However, traditional binarization techniques (Rastegari et al., 2016) often treat weights as independent entities, failing to exploit the inherent structure and patterns in the weight matrices. These observations encourage us to explore N:M sparsity tailored specifically for 1-bit LLMs to achieve further speedups and compression gains.

Based on these observations, we develop our STBLLM approach, STructured Binarization for LLMs to achieve extreme compression while mitigating performance degradation. To our knowledge, STBLLM is the first to push LLM compression below 1-bit precision, opening new possibilities for edge deployment. Our workflow applies the metric-based sparsity and performs the adaptive N:M binarization. Specifically, to measure the importance of weights, we introduce a Standardized Importance (SI) metric that addresses the issues of extreme weight values and computationally expensive Hessian-based methods used in prior work. We then propose an adaptive layer-wise struc-

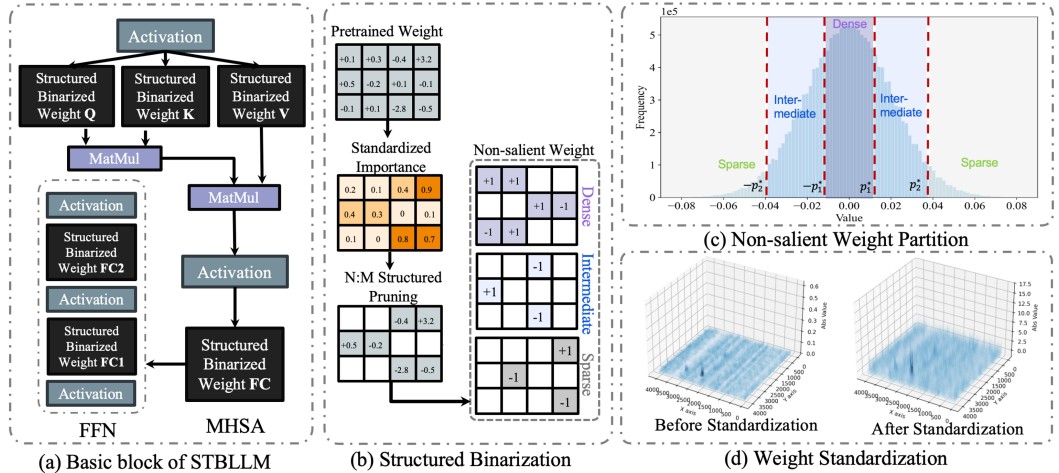

Figure 3: (a) PTQ framework in Structured Binarized LLM (STBLLM). We apply structured binarization to all of the weights. (b) Structured Binarized Weight Computation Procedure. We first perform N:M structure pruning to pre-trained weight (here N=2, M=4), then show how we process the non-salient weight. (c) Trisection partition for Symmetric Gaussian Distribution of Non-salient Weight. (d) Illustration of Weight Standardization on LLaMA-2-7B.

tured binarization approach, where different layers of the LLM can be sparsified with varying N:M ratios to balance compression and accuracy. We employ a residual approximation technique (Huang et al., 2024) for the salient parameters to preserve the critical information. For the non-salient parameters, we utilize a fine-grained grouping strategy based on a trisection search algorithm to find optimal splitting points $p^*$ and apply different quantization schemes to the sparse, intermediate, and dense regions as presented in Figure 3(c). By tailoring these structured representations specifically for 1-bit LLMs, we unlock a new avenue for model compression and optimization, enabling more widespread deployment of these powerful LLMs in resource-constrained environments.

To validate the effectiveness of STBLLM, we conduct extensive experiments on various LLMs, including the LLaMA-1/2/3 (Touvron et al., 2023a;b), OPT (Zhang et al., 2022a) and Mistral (Jiang et al., 2023). As presented in Figure 2, our STBLLM achieves a better trade-off between performance and bit-width. STBLLM with 0.8 bit can achieve lower perplexity than BiLLM with 1.1 bit. STBLLM achieves a perplexity of 31.72 at just 0.55 bits per weight, compared to 688.73 for BiLLM - an over 20× gain. Even at 65B parameters, our 0.55-bit STBLLM outperforms BiLLM's 0.7-bit and PB-LLM's 1.7-bit versions. STBLLM retains significantly higher accuracy for zero-shot benchmarks than BiLLM under 4:8 and 6:8 structured binarization settings across 13B and 30B LLaMA. For example, on LLaMA-1-30B, our 0.55-bit STBLLM achieves 51.78% average accuracy versus just 43.72% for BiLLM. The contribution of our work is as follows:

- We introduce STBLLM, a novel structural binarization framework that compresses Large Language Models (LLMs) to less than 1-bit precision, enabling significant memory and computational savings while preserving model performance.

- STBLLM employs an N:M binary weight kernel approach, where we perform structural binarization of the weights using efficient gradient-free metrics to determine weight importance, channel rearrangement to preserve salient weights, and adaptive layer mixed-structure binarization for better accuracy-efficiency trade-offs.

- We implement a specialized CUDA kernel for structural binarization, leveraging NVIDIA's Ampere GPU sparse tensor cores, achieving a 17.85x speedup over ABQ-LLM's 2-bit implementation.

- Extensive experiments on LLMs like LLaMA and OPT show that STBLLM outperforms other compressed binarization methods, achieving a perplexity of 11.07 at 0.55 bits per weight—3× better than BiLLM.

## 2 RELATED WORK

**Quantization and Binarization.** Quantization reduces full-precision parameters to lower bits, thereby decreasing storage and computation requirements. Recent research has effectively applied Quantization-Aware Training (QAT) and Post-Training Quantization (PTQ) to LLMs. QAT (Liu et al., 2024; Du et al., 2024; Chen et al., 2024) incorporates quantization during training, allowing to learn better representations for low-bit weights. However, due to the massive parameters, retraining is too costly and inefficient for LLMs. In contrast, PTQ (Frantar et al., 2023; Chee et al., 2023; Lin et al., 2024b; Lee et al., 2024; Dong et al., 2023b) can directly quantize pre-trained models without additional training. GPTQ (Frantar et al., 2023) and QuIP (Chee et al., 2023) minimized LLM block quantization errors through second-order error compensation. Other approaches (Lin et al., 2024b; Lee et al., 2024; Dettmers et al., 2023) focus on prioritizing salient weights to maintain their information representation capacity. Binarization, which constrains quantized parameters to a 1-bit representation, is the most extreme quantization method and has proven effective for vision tasks, such as XNOR-Net (Rastegari et al., 2016) and Bi-Real Net (Liu et al., 2018). To further compress binary neural networks, sparse kernel techniques (Wang et al., 2020; 2021b;a; Liu et al., 2022) are introduced to reduce the redundancy in binary neural networks. For LLM binarization, BitNet (Wang et al., 2023a) trained 1-bit LLM from scratch. OneBit (Xu et al., 2024) employ the QAT paradigm for 1-bit LLM while BiLLM (Huang et al., 2024) employ the PTQ paradigm with residual approximation technique. In this paper, we further reduce weights below 1-bit by identifying and removing the redundant parameters.

**Sparsity Methods for LLM.** Pruning removes less important parameters from a neural network to reduce its size and improve efficiency. For LLMs, Pruning can be divided to structured pruning (Ma et al., 2023; Ashkboos et al., 2024; Xia et al., 2024; An et al., 2023), semi-structured pruning (Frantar & Alistarh, 2023; Sun et al., 2024; Zhang et al., 2024b) and unstructured pruning (Li et al., 2024b;d; Dong et al., 2024). Structured pruning methods, including LLM-Pruner (Ma et al., 2023) and Sheared LLaMA (Xia et al., 2024), aim to simplify LLM by removing specific components such as heads, layers, and dimensions. Although these techniques enhance model efficiency, they often result in significant performance degradation and require extensive retraining to recover lost capabilities. In contrast, unstructured pruning methods (Frantar & Alistarh, 2023; Sun et al., 2024) remove individual weights based on their significance within the model. However, this approach leads to irregular sparsity patterns that do not effectively leverage hardware acceleration. Semi-structured pruning offers a more balanced approach to model optimization. Methods such as SparseGPT (Frantar & Alistarh, 2023) and Wanda (Sun et al., 2024) exemplify this strategy by maintaining regular, hardware-friendly sparsity patterns, such as N:M sparsity. This approach combines the fine-grained control characteristic of unstructured pruning with the operational efficiency associated with structured pruning.

**Synergy of Pruning and Quantization.** Research highlights the complementary strengths of pruning and quantization. Pruning trims neural network parameters, while quantization compresses their precision. Deep Compression (Han et al., 2016) integrates pruning, trained quantization, and Huffman coding into a streamlined framework, significantly lowering the storage footprint of deep neural networks (DNNs). Later works (Tung & Mori, 2018; Yang et al., 2019; Hu et al., 2021) develop in-parallel pruning-quantization methods to optimize compression allocation, such as unstructured pruning sparsity and quantization strategies. For extreme cases like binarization, approaches like STQ-Nets (Munagala et al., 2020) (structured pruning for CNNs), BNN Pruning (Li & Ren, 2020) (weight flipping-based pruning), and BAP (Wang et al., 2021a) (98% sparsity via sparse convolution) achieve significant compression and speedup. However, their reliance on fine-tuning limits applicability to LLMs.

## 3 METHODOLOGY

In this section, we introduce our STBLLM framework, as depicted in Figure 3. We employ structured binarization for all weights within the Feed-forward Network (FFN) and Multi-head Self-attention (MHSA) modules. Specifically, we introduce the concept of Standardized Importance (SI) to evaluate the saliency of each weight under N:M sparsity constraints (refer to the left part of Figure 3(b)). We leverage the Hessian matrix to distinguish between salient and non-salient weights for the binarization process. Salient weights are handled using residual approximation, following the

---

**Algorithm 1** Framework of STBLLM: Details of each function are shown in Algorithm 2.

---

1: **function** STRUCTUREDBINARYLLM($\mathbf{W}$, $\mathbf{X}$, $\beta$, $\lambda$)
2:     **Input:** $\mathbf{W} \in \mathbb{R}^{n \times m}$ denotes weight matrix; $\mathbf{X} \in \mathbb{R}^{r \times d}$ represents calibration data;
3:     $\beta$ denotes block size; $\lambda$ represents hessian regularizer
4:     **Output: B** - structured binarized weights
5:     $\mathbf{H} \leftarrow 2\mathbf{X}\mathbf{X}^{\top}$          $\triangleright$ $\ell^2$ error hessian matrix
6:     $\mathbf{H}^c \leftarrow \text{Cholesky}((\mathbf{H} + \lambda\mathbf{I})^{-1})$
7:     $\mathbf{B} \leftarrow 0_{n \times m}$
8:     **for** $b = 0, \beta, 2\beta, ..., N$ **do**
9:         $\mathbf{W}^{si} \leftarrow \text{Standardized\_Importance}(\mathbf{W}_{:,b:b+\beta})$
10:        $\mathbf{W}^s \leftarrow \text{Semi-Structured}(\mathbf{W}^{si}_{:,b:b+\beta}, \mathbf{W}_{:,b:b+\beta})$
11:        $row_s\{\cdot\} \leftarrow \text{Salient}(\mathbf{W}_{:,b:b+\beta}, \mathbf{H}^c)$
12:        $\tilde{\mathbf{B}}_1 \leftarrow \text{Res\_Approx}(\mathbf{W}^s_{:,j \in \{row_s\}})$
13:        $p_1^*, p_2^* \leftarrow \text{NonSalientAwareQuant}(\mathbf{W}^s_{i,j \notin \{row_s\}})$
14:        $\tilde{\mathbf{B}}_2, \tilde{\mathbf{B}}_3, \tilde{\mathbf{B}}_4 \leftarrow \text{Trisection}(\mathbf{W}_{|w_{i,j}|}, p_1^*, p_2^*)$
15:        $\mathbf{B}_{:,b:b+\beta} \leftarrow \tilde{\mathbf{B}}_1 \cup \tilde{\mathbf{B}}_2 \cup \tilde{\mathbf{B}}_3 \cup \tilde{\mathbf{B}}_4$
16:        $\mathbf{E} \leftarrow (\mathbf{W}_{:,b:b+\beta} - \mathbf{B}_{:,b:b+\beta})/\mathbf{H}^c_{b:b+\beta,b:b+\beta}$
17:        $\mathbf{W}_{:,b+\beta:} \leftarrow \mathbf{W}_{:,b+\beta:} - \mathbf{E} \cdot \mathbf{H}^c_{b:b+\beta,b+\beta:}$      $\triangleright$ block-wise OBC
18:     **end for**
19:     **return** B
20: **end function**

---

methodology outlined in BiLLM (Huang et al., 2024). For non-salient weights, we propose a Non-salient Aware Quantization technique, which further divides these weights into Dense, Intermediate, and Sparse regions (as shown in the right part of Figure 3(c)). To optimally partition the non-salient weights into three distinct regions, we utilize a trisection search strategy to determine the appropriate $p_1^*$ and $p_2^*$ values. In the subsequent update step, we apply block-wise error compensation (Frantar & Alistarh, 2023; Frantar et al., 2023) to preserve performance following post-training quantization (PTQ). Algorithm 1 summarizes the STBLLM process, with implementation details in Appendix A. Building upon BiLLM (Huang et al., 2024), we refine the SI metric, adopt semi-structured adaptive pruning, and implement Non-Salient aware quantization to distinguish our approach.

## 3.1 PRELIMINARIES

**Binarization** quantizes floating-point (FP) weights, represented as $\mathbf{W}_{FP}$, into 1-bit values (i.e., $\pm 1$). During forward propagation, the sign function is used to binarize the original parameter tensor:

$$\mathbf{B} := \alpha \cdot \text{sign}(\mathbf{W}_{FP}), \tag{1}$$

$$\text{sign}(w) := \begin{cases} 1 & \text{if } x \geq 0, \\ -1 & \text{others,} \end{cases} \tag{2}$$

where $\mathbf{W}_{FP} \in \mathbb{R}^{n \times m}$ is the 32-bit floating-point weight, and $\mathbf{B} \in \mathbb{R}^{n \times m}$ is the binarized output, and $\alpha := \frac{||\mathbf{W}||_{l_1}}{m}$. The parameter $n$ and $m$ represent the size of the weight matrix. The scaling factor $\alpha \in \mathbb{R}^n$ is applied in a channel-wise manner (Rastegari et al., 2016).

**N:M Sparsity.** Inspired by the experiments in Figure 1, we observe the binarized the redundancy in LLMs. By applying the N:M binarization for LLMs, we can achieve an extreme compression ratio of less than 1 bit. Specifically, we introduce a N:M sparsity that encodes N consecutive non-zero elements in the weight matrix using a M-bit representation. While it enhances acceleration, it compromises performance. To alleviate this problem, we propose several techniques from different perspectives: ① **Importance Measurement.** Previous methods (Frantar et al., 2023; Chen et al., 2024; Huang et al., 2024) utilize Hessian-based methods to measure the importance, but they are computationally expensive and can not reflect the true importance of parameters in LLMs. ② **Layer-wise Assignment.** Previous PTP methods (Frantar & Alistarh, 2023; Zhang et al., 2024b) utilize the uniform sparsity ratio among different layers. However, recently, evidence (Yin et al., 2024) shows that not all layers have the same redundant level thus non-uniform sampling can help retain the

performance. ③ **Hierarchical Quantization.** Prior PTQ techniques for LLMs (Lin et al., 2024b; Lee et al., 2024; Huang et al., 2024) separate weights into salient and non-salient categories via activation magnitude or Hessian matrix, focusing on salient weights for their role in performance. Here, we reevaluate non-salient parameters, underscoring their vital influence on quantization.

## 3.2 STANDARDIZED IMPORTANCE METRIC

Many previous works, such as DB-LLM (Chen et al., 2024), SparseGPT (Frantar & Alistarh, 2023), GPTQ (Frantar et al., 2023), and BiLLM (Huang et al., 2024), utilize the Hessian metric to measure the importance of weights. However, we observe that extreme values in the weights significantly impact Hessian computation (See Appendix D). To address this issue, we present a Standardized Importance (SI) metric. The computation of SI does not involve the second-order information of the weights, which can be computationally expensive for LLMs. Specifically, we employ standardization to mitigate the issue of extreme values in weights by transforming the weights to have a mean of zero and a standard deviation of one. This process ensures that all weights are on a similar scale, reducing the disproportionate influence of extreme values on the Hessian matrix. For a linear layer with weight $\mathbf{W} \in \mathbb{R}^{n \times m}$, which takes in input activation $\mathbf{X} \in \mathbb{R}^{r \times d}$, where $r$ is the batch size and $d = m$ is the input dimension. We evaluate the importance of each weight by the product of its magnitude and the corresponding input feature norm. The score for the $\mathbf{W}_{i,j}$ is defined as:

$$\mathbf{S}_{i,j} = \sigma(\mu(|\mathbf{W}_{i,j}|)) \cdot ||\mathbf{X}_{:,j}||_2, \quad \sigma(w) = \frac{w - \mu_{\mathbf{W}}}{\sigma_{\mathbf{W}}}, \quad \mu(|\mathbf{W}_{i,j}|) = \frac{|\mathbf{W}_{i,j}|}{\sum_j |\mathbf{W}_{i,j}|} + \frac{|\mathbf{W}_{i,j}|}{\sum_i |\mathbf{W}_{i,j}|}, \quad (3)$$

where $\sigma(\cdot)$ standardizes the weight magnitude $\mu(|\mathbf{W}_{i,j}|)$ using the mean $\mu_{\mathbf{W}}$ and standard deviation $\sigma_{\mathbf{W}}$ of all weights in the layer. The weight magnitude $\mu(|\mathbf{W}_{i,j}|)$ is computed as the sum of the L1-normalized magnitude across the input dimension $j$ and the output dimension $i$. The input feature norm $||\mathbf{X}_{:,j}||_2$ is calculated as the L2 norm of the $j$-th column input activation $\mathbf{X}$. By multiplying the standardized weight magnitude $\sigma(\mu(|\mathbf{W}_{i,j}|))$ with the input feature norm $||\mathbf{X}_{:,j}||_2$, the importance score $\mathbf{S}_{i,j}$ takes into account both the significance of the weight itself and the activation level of the associated input feature. To prune the linear layer, we rank all the weights based on their importance scores $\mathbf{S}_{i,j}$ and remove a specified percentage of the weights with the lowest scores. This pruning strategy aims to preserve the most significant weights contributing to the layer's output while eliminating less important weights to reduce the model's size and computational complexity.

## 3.3 ADAPTIVE LAYER-WISE BINARIZATION

**N:M Binary Weight Vector.** To achieve compression beyond standard binarization, we propose an $N : M$ sparsity approach, where M binary values are represented by N values (N < M). This allows for further compression while preserving the salient information in the weight tensors. Specifically, we employ the mixed N:8 sparsity configuration following DominoSearch (Sun et al., 2021).

**Layer-wise N:M Assignment.** To achieve better accuracy-efficiency trade-offs, we introduce adaptive layer-wise structured binarization, where different layers of the LLM can be sparsified with different N:M ratios. (For example, with a target ratio of 4:8, layers can have ratios like 3:8, 4:8, and 5:8 while maintaining the overall 4:8 ratio.) This flexibility allows for more aggressive compression in less important layers while preserving higher precision in crucial layers. The layer-wise N:M ratios are assigned based on the relative importance of each layer, measured by the L2 norm of its weight parameters. Let $\omega_i$ and $\omega_{\text{total}}$ be the L2 norm of layer $i$ and the sum across all layers, respectively. The relative importance $\alpha_i$ of layer $i$ is $\alpha_i = \frac{\omega_i}{\omega_{\text{total}}}$. The N:M ratio for layer $i$ is $\frac{N_i}{M_i} = \alpha_i + (1 - \alpha_i) \cdot R_{\text{target}}$, where $R_{\text{target}}$ is the target overall compression ratio. More important layers have higher N:M ratios (less sparsification), approaching 1:1 for the most important ones. Less important layers have lower N:M ratios, approaching $R_{\text{target}}$ for the least important ones. This ensures the overall compression ratio meets $R_{\text{target}}$.

## 3.4 NON-SALIENT AWARE QUANTIZATION

Based on the observations that a small fraction of salient weights is critical to the LLM quantization (Lin et al., 2024b; Shao et al., 2023), we split the weights into the salient and non-salient parts and then apply a higher bit for salient one and lower-bit for non-salient one, as:

Table 1: Average bit results from structural searching and residual binarization of OPT, LLaMA-1, and LLaMA-2 families. *Results for largest models: OPT-66B, LLaMA-1-65B, LLaMA-2-70B.

| Model | BiLLM | | | | BiLLM-4:8 | | | | BiLLM-5:8 | | | | BiLLM-6:8 | | | |
|---|---|---|---|---|---|---|---|---|---|---|---|---|---|---|---|---|
| | 7B | 13B | 30B | 65-70B* | 7B | 13B | 30B | 65-70B* | 7B | 13B | 30B | 65-70B* | 7B | 13B | 30B | 65-70B* |
| OPT | 1.10 | 1.12 | 1.12 | 1.13 | 0.55 | 0.56 | 0.56 | 0.56 | 0.69 | 0.70 | 0.70 | 0.71 | 0.83 | 0.84 | 0.84 | 0.85 |
| LLaMA-1 | 1.09 | 1.09 | 1.10 | 1.10 | 0.54 | 0.54 | 0.55 | 0.55 | 0.68 | 0.68 | 0.69 | 0.69 | 0.82 | 0.82 | 0.83 | 0.83 |
| LLaMA-2 | 1.07 | 1.08 | N/A | 1.09 | 0.53 | 0.54 | N/A | 0.54 | 0.67 | 0.67 | N/A | 0.68 | 0.80 | 0.81 | N/A | 0.82 |

Table 2: Perplexity comparison of PB-LLM and BiLLM on the LLaMA model family. The columns represent the perplexity results on the Wikitext2 for different model sizes. The average bit-width for each model is provided in the table. For more precise bit-width results, please refer to Table 1.

| Settings | | | LLaMA-1 | | | | LLaMA-2 | | LLaMA-3 |
|---|---|---|---|---|---|---|---|---|---|
| Method | Block Size | W-Bits | 7B | 13B | 30B | 65B | 7B | 13B | 8B |
| FullPrecision | - | 16 | 5.68 | 5.09 | 4.1 | 3.53 | 5.47 | 4.88 | 6.10 |
| RTN | - | 1 | 1.7e5 | 1.4e6 | 1.5e4 | 6.5e4 | 1.6e5 | 4.8e4 | 2.7e6 |
| GPTQ | 128 | 1 | 2.7e5 | 1.1e5 | 6.7e4 | 2.5e4 | 1.2e5 | 9.4e3 | 5.7e4 |
| PB-LLM | 128 | 1.7 | 102.36 | 36.6 | 33.67 | 12.53 | 69.2 | 151.09 | 41.80 |
| BiLLM | 128 | 1.09 | 35.04 | 15.14 | 10.52 | 8.49 | 32.48 | 16.77 | 28.30 |
| BiLLM | 128 | 0.80 (6:8) | 80.36 | 22.55 | 13.22 | 9.09 | 50.25 | 27.28 | 94.15 |
| BiLLM | 128 | 0.70 (5:8) | 126.99 | 39.61 | 18.69 | 11.57 | 87.84 | 58.14 | 161.48 |
| BiLLM | 128 | 0.55 (4:8) | 688.73 | 124.72 | 37.96 | 29.22 | 263.61 | 124.78 | 663.91 |
| STBLLM | 128 | 0.80 (6:8) | 15.03 | 9.66 | 7.56 | 6.43 | 13.06 | 11.67 | 33.44 |
| STBLLM | 128 | 0.70 (5:8) | 19.48 | 11.33 | 9.19 | 7.91 | 18.74 | 13.26 | 49.12 |
| STBLLM | 128 | 0.55 (4:8) | 31.72 | 17.22 | 13.43 | 11.07 | 27.93 | 20.57 | 253.76 |

**Salient Part:** In our cases, for salient weight, we apply residual approximation (Huang et al., 2024), which is composed of residual approximation weight, as follows:

$$\begin{cases} \alpha_o^*, \mathbf{B}_o^* = \arg\min_{\alpha_o, \mathbf{B}_o} \|\mathbf{W} - \alpha_o \mathbf{B}_o\|^2, \\ \alpha_r^*, \mathbf{B}_r^* = \arg\min_{\alpha_r, \mathbf{B}_r} \|(\mathbf{W} - \alpha_o^* \mathbf{B}_o^*) - \alpha_r \mathbf{B}_r\|^2, \end{cases} \tag{4}$$

where $\mathbf{B}_o$ denotes the original binary tensor, and $\mathbf{B}_r$ represent the residual binarized matrix as the compensation. The final approximation of $\mathbf{W}$ is $\mathbf{W} \approx \alpha_o^* \mathbf{B}_o^* + \alpha_r^* \mathbf{B}_r^*$.

**Non-Salient Part:** For the non-salient part (which is also symmetric Gaussian distribution), we find that significant information is retained in the non-salient part. To make the trade-off with bit and performance, we utilize a fine-grained grouping strategy called the Trisection search algorithm (See Algorithm 2), a divide-and-conquer method to find optimal split points $p_1^*$ and $p_2^*$ for weight regions. With these two break-points, we can segment the symmetric Gaussian distribution into three groups, which is sparse $R_s[-m, -p_2^*] \cup [p_2^*, m]$, intermediate $R_i[-p_2^*, -p_1^*] \cup [p_1^*, p_2^*]$, and dense region $R_d[-p_1^*, p_1^*]$. Then, we derive the quantization error:

$$\theta_{p_1^*, p_2^*}^2 = \|\mathbf{W}_s - \alpha_s \mathbf{B}_s\|^2 + \|\mathbf{W}_i - \alpha_i \mathbf{B}_i\|^2 + \|\mathbf{W}_d - \alpha_d \mathbf{B}_d\|^2, \tag{5}$$

$$\alpha_s = \frac{1}{n_s}\|\mathbf{W}_s\|_{l1}, \ \ \alpha_i = \frac{1}{n_i}\|\mathbf{W}_i\|_{l1}, \ \ \alpha_d = \frac{1}{n_d}\|\mathbf{W}_d\|_{l1} \tag{6}$$

where $\mathbf{W}_s$, $\mathbf{W}_i$, $\mathbf{W}_d$ are the sums of absolute weight values in the sparse, intermediate, and dense regions. $\mathbf{B}_s$, $\mathbf{B}_i$, $\mathbf{B}_d$ are the binarized weights for those regions. These three regions are binarized separately. This method introduces an additional 2 bits for group identification, which constitutes a minor portion of the overall bit count, while the majority of computing parameters remain at 1 bit.

**Average Bits.** In STBLLM, we introduce extra bits while pruning the redundant or less important weights. The overhead of weight parameters is $N_{\text{param}} = 2 \times r_{\text{salient}} + 1 \times (1 - r_{\text{salient}})$. The additional hardware overhead is $N_{\text{storing}} = 2 + \frac{1}{b_{\text{size}}}$, where $r_{salient}$ denotes the proportion of salient weights and $b_{\text{size}}$ denotes the block size in OBC compensation, with 2 bits allocated for marking the division of non-salient weights. Under N:M binarization settings, where N and M are positive integers with $N < M$, we prune the model weights by retaining only a fraction (N/M) of the original weights. Consequently, the number of parameters in the pruned STBLLM model is $N_{\text{stbllm}} = N_{\text{param}} \times \frac{N}{M}$. This N:M binarization method allows for a significant reduction in model size.

Table 3: Perplexity results on Wikitext2 datasets of OPT and Mistral models with BiLLM and STBLLM. For more precise bit-width results, please refer to Table 1.

| Settings | | OPT | | | | Mistral |
|---|---|---|---|---|---|---|
| Method | W-Bits | 1.3B | 2.7B | 6.7B | 30B | 7B |
| BiLLM | 0.80 (6:8) | 51.62 | 23.03 | 15.82 | 15.82 | 72.29 |
| BiLLM | 0.70 (5:8) | 69.15 | 30.62 | 20.58 | 20.58 | 82.84 |
| BiLLM | 0.55 (4:8) | 106.99 | 55.28 | 79.68 | 79.68 | 189.73 |
| STBLLM | 0.80 (6:8) | 29.84 | 17.02 | 12.79 | 12.80 | 27.31 |
| STBLLM | 0.70 (5:8) | 33.01 | 20.82 | 14.38 | 14.38 | 25.64 |
| STBLLM | 0.55 (4:8) | 45.11 | 30.34 | 18.80 | 18.80 | 70.14 |

Table 4: Accuracies (%) for 7 zero-shot tasks from structured binarized LLaMA-1-13B, LLaMA-2-13B, and LLaMA-1-30B with BiLLM and STBLLM. We compare the performance under the same N:M setting to achieve sub-1-bit quantization.

| Models | Method | Winogrande | OBQA | Hellaswag | Boolq | ARC-e | ARC-c | RTE | Mean |
|---|---|---|---|---|---|---|---|---|---|
| LLaMA-1-13B | FullPrecision | 72.77 | 33.20 | 59.94 | 77.89 | 77.40 | 46.50 | 70.40 | 62.59 |
| | BiLLM(6:8) | 58.80 | 30.60 | 46.25 | 62.96 | 49.96 | 23.97 | 53.42 | 46.57 |
| | BiLLM(4:8) | 52.09 | 28.00 | 30.82 | 61.25 | 32.66 | 21.25 | 53.07 | 39.88 |
| | STBLLM(6:8) | 65.98 | 36.20 | 63.67 | 65.38 | 68.86 | 34.04 | 56.68 | 55.83 |
| | STBLLM(4:8) | 63.06 | 34.80 | 52.65 | 62.48 | 56.90 | 28.33 | 52.71 | 50.13 |
| LLaMA-2-13B | FullPrecision | 72.22 | 35.20 | 60.06 | 80.52 | 79.42 | 48.46 | 65.34 | 63.03 |
| | BiLLM(6:8) | 56.43 | 30.60 | 35.53 | 62.48 | 41.29 | 24.74 | 53.43 | 43.50 |
| | BiLLM(4:8) | 50.59 | 24.00 | 28.96 | 62.08 | 30.51 | 22.35 | 53.07 | 38.79 |
| | STBLLM(6:8) | 63.93 | 37.00 | 57.76 | 71.53 | 60.56 | 31.99 | 54.15 | 53.85 |
| | STBLLM(4:8) | 55.88 | 29.40 | 44.03 | 64.31 | 48.86 | 26.54 | 52.71 | 45.96 |
| LLaMA-1-30B | FullPrecision | 75.69 | 36.00 | 63.35 | 82.69 | 80.30 | 52.82 | 66.79 | 65.38 |
| | BiLLM(6:8) | 66.54 | 36.40 | 58.18 | 66.15 | 62.37 | 31.91 | 46.93 | 50.32 |
| | BiLLM(4:8) | 54.93 | 29.40 | 38.85 | 62.17 | 43.6 | 24.74 | 52.35 | 43.72 |
| | STBLLM(6:8) | 71.59 | 41.00 | 69.85 | 77.37 | 71.55 | 41.3 | 48.01 | 60.10 |
| | STBLLM(4:8) | 64.01 | 34.60 | 56.46 | 63.06 | 60.86 | 31.48 | 51.99 | 51.78 |

# 4 EXPERIMENTS

## 4.1 IMPLEMENTATION DETAILS

**Experimental Setup.** Our STBLLM utilizes PyTorch (Paszke et al., 2019) and Huggingface (Wolf et al., 2019) libraries. Most LLMs except 65B can be evaluated on a single NVIDIA A800 GPU. For the LLaMA-1-65B model, we employ four NVIDIA A800 GPUs for evaluation. It takes 1.8 hours for the post-training process of 7B models on an RTX 4090 GPU and 2.8 hours for 13B models on an A6000 GPU. Following BiLLM (Huang et al., 2024), our proposed STBLLM also eliminates the need for fine-tuning, offering an efficient post-training quantization framework.

**Datasets and Models.** We measure the perplexity for language generation tasks on Wikitext2 (Merity et al., 2016), C4 (Raffel et al., 2020) and PTB (Marcus et al., 1993), and accuracy for the zero-shot tasks including Winogrande (Sakaguchi et al., 2021), OBQA (Mihaylov et al., 2018), Hellaswag (Zellers et al., 2019), BoolQ (Clark et al., 2019), ARC (Clark et al., 2018) and RTE (Chakrabarty et al., 2021). We conduct experiments on LLaMA-1/2/3 (Touvron et al., 2023a;b), OPT (Zhang et al., 2022a), and Mistral (Jiang et al., 2023). For perplexity evaluation in Table 2 and 3, we employ the C4 dataset as the calibration dataset and report the perplexity on Wikitext2.

**Baseline.** Our primary baseline is BiLLM (Huang et al., 2024), which is a 1-bit PTQ framework for LLMs. We perform an N:M sparse pattern on pre-trained LLMs and then conduct the same procedure as BiLLM to report the results that are less than 1 bit (e.g. 0.8, 0.7, 0.55 bits). We conduct the N:M sparsity using Wanda (Sun et al., 2024) as the baseline, a gradient-free post-training pruning method. We compare the results of STBLLM with BiLLM under the same N:M settings. For more information on average bits under N:M settings, please refer to Table 1. Previous low-bit methods like PB-LLM (Shang et al., 2024), GTPQ (Frantar et al., 2023) and RTN are selected for comparison.

## 4.2 MAIN RESULTS

**Comparison with PTQ methods.** We comprehensively compare the performance of different LLaMA families across various model sizes (7B-65B). For a fair comparison, we set the same block size to 128. As presented in Table 2, the model under RTN and GPTQ fails to retain the performance at 1-bit. PB-LLM has shown a satisfactory perplexity under 1.7 bit but deteriorates performance compared with BiLLM under 1.09 bit. To further compare the performance at sub-1-bit, we apply the same N:M setting to BiLLM and our proposed STBLLM. As shown in Figure 2, our proposed STBLLM achieves a better trade-off between bit-widths and perplexity across model sizes from 7B to 65B. STBLLM surpasses BiLLM by a large margin ($688.73 \rightarrow 31.72$) on LLaMA-1-7B, especially on the most extreme compression case, 4:8 structured binarization, which means setting half of the parameter to zero. It is also noteworthy that when the parameter size reaches 65B, our STBLLM, at 0.55 bit, achieves a perplexity of 11.07, surpassing that of PB-LLM (12.53) at 1.7 bit and that of BiLLM (11.57) at 0.7 bit. To our knowledge, our STBLLM is the first work that breaks the 1-bit barriers by further reducing the redundant weights in an N:M pattern. Moreover, we conduct further experiments on the OPT family from 1.3B to 30B and Mistral-7B at sub-1-bit PTQ settings. From Table 3, we observe the same trend as in LLaMA. Our proposed STBLLM performs significantly better than BiLLM across all models and all N:M structured binarization settings.

Table 5: Ablation for pruning metric.    Table 6: Ablation study for allocation strategy.

| Model | Magnitude | Wanda | SparseGPT | Ours (SI) | | Models | Uniform | Sin-shape | Ours |
|---|---|---|---|---|---|---|---|---|---|
| LLaMA-1-7B | 4797.41 | 207.32 | 32.82 | 31.72 | | LLaMA-1-7B | 80.36 | 67.78 | 15.03 |
| LLaMA-2-7B | 2287.24 | 97.54 | 31.55 | 27.93 | | LLaMA-2-7B | 50.25 | 33.61 | 13.06 |

**Zero-Shot Performance.** To conduct a more comprehensive evaluation of binary LLMs, we extend our experiments to 7 zero-shot datasets on LLaMA-1-13B, LLaMA-2-13B, and LLaMA-1-30B, each tested with FullPrecision, BiLLM(6:8), BiLLM(4:8), STBLLM(6:8), and STBLLM(4:8) methods. We mainly focus on the performance of these models under the sub-1-bit setting. Specifically, we compare the BiLLM and our STBLLM under 4:8 and 6:8 structured binarization settings. As illustrated in Table 4, we find that the performance drop in reduced precision methods is more pronounced in BiLLM methods compared to STBLLM methods, indicating that STBLLM methods are more robust alternatives when memory resources are constrained.

## 4.3 ENHANCING INFERENCE EFFICIENCY ON HARDWARE.

We present specialized CUDA kernels designed to support 1-bit 2:4 sparsification. As illustrated in Figure 4(a), we utilize a 2-bit implementation on recent RTX4090 GPU from ABQ-LLM (Zeng et al., 2024) as the baseline (W2A16 and W2A8) and compare it with our highly-optimized 2:4 1-bit implementation. We provide a comparative analysis of runtime and throughput across various sequence lengths, demonstrating the significant gains in computational efficiency and reduced memory footprint. Specifically, for typical sequence lengths of 4096 and 8192, our implementation achieves up to 17.85 times speedup compared to ABQ-LLM's 2-bit implementation. This speedup stems from our CUDA kernel's efficient use of Sparse Tensor Cores, detailed in Appendix C. At a sequence length of 8192, our kernel reaches 263.45 TFLOPS, which is 79.74% of the RTX4090's 2:4 sparse tensor core peak performance. Notably, the speedup becomes more pronounced as sequence length increases. Furthermore, as illustrated in Figure 4 (b), our method yields lower perplexity for LLaMA-1/2 models. This indicates superior model performance and accuracy compared to 2-bit round-to-nearest (RTN), GPTQ, and AWQ. Refer to Appendix C for the memory comparison and implementation details.

## 4.4 ABLATION STUDIES

**Ablation for Metric.** Table 5 shows the impact of post-training pruning metrics (Magnitude, Wanda (Sun et al., 2024), SparseGPT (Frantar & Alistarh, 2023) and our SI) on 4:8 binary STBLLM regarding LLaMA-1/2-7B. During PTP, we employ the C4 dataset as the calibration dataset and report the perplexity on the Wikitext2 dataset. SparseGPT requires second-order information, which involves a massive computation burden. Similar to Wanda, our SI does not require gradient or second-order information. Our method achieves better performance among these metrics.

Table 7: Comparison of Magnitude, Wanda, SparseGPT, and SI across different datasets.

| Models | LLaMA-1-7B | | | | Models | LLaMA-2-7B | | | |
|---|---|---|---|---|---|---|---|---|---|
| Dataset | Magnitude | Wanda | SparseGPT | Ours(SI) | Dataset | Magnitude | Wanda | SparseGPT | Ours(SI) |
| PTB | 11608.88 | 306.57 | 61.53 | 68.48 | PTB | 45564.36 | 2027.33 | 236.03 | 690.76 |
| C4 | 1545.34 | 153.29 | 33.06 | 36.04 | C4 | 1034.84 | 86.45 | 30.53 | 30.81 |
| Wikitext2 | 4797.42 | 207.32 | 32.82 | 31.72 | Wikitext2 | 2287.25 | 97.54 | 31.56 | 27.93 |

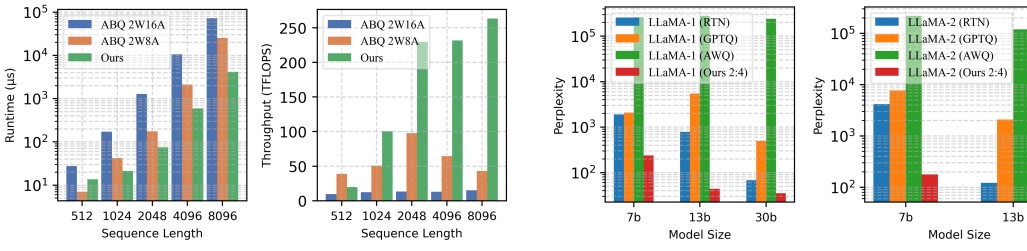

(a) Runtime and throughput comparison.      (b) Perplexity comparison.

Figure 4: (a) Runtime and throughput comparison across sequence lengths for ours and ABQ-LLM. (b) Perplexity comparison across model sizes under 2:4 setting for LLaMA-1/2.

**Ablation for Quantization Strategy.** We conduct an ablation study on different quantization strategies. Comparing the perplexity of our Non-salient Aware Quantization (dubbed as Non-salient) and Bell-shaped Distribution Splitting (dubbed as Bell-shaped) in BiLLM (Huang et al., 2024) on both LLaMA-1-7B and LLaMA-2-7B, as shown in Table 8. The perplexity of Non-salient changes a lot when moving from LLaMA-1-7B to LLaMA-2-7B, while our Non-salient exhibits nearly identical perplexity in both models, significantly lower than that of Bell-shaped.

**Ablation for Allocation Strategy.** Table 6 presents an ablation study on different allocation strategies. We compare our method with Uniform and Sin-shaped allocation strategies. The Sin-shaped strategy assigns layer-wise sparsity following a sine wave pattern, where the initial layers have lower sparsity and the latter have higher sparsity. The performance of Uniform and Sin-shaped strategies varies significantly across different models. In contrast, our strategy consistently achieves nearly identical performance across both models, outperforming the other two allocation strategies.

**Ablation for Group Size.** Table 9 presents the results of our ablation study on the group size configuration. We evaluate the perplexity of LLaMA-1-7B and LLaMA-2-7B with group sizes of 64, 128, 256, and 512. Generally, as the group size increases, performance improves. However, this also results in higher computational and storage demands. We choose a group size of 128 to balance performance and resource consumption.

Table 8: Ablation for quantization strategy.      Table 9: Ablation for group size.

| Models | Bell-shaped | Non-salient |
|---|---|---|
| LLaMA-1-7B | 80.35 | 15.03 |
| LLaMA-2-7B | 50.25 | 13.06 |

| Model | 64 | 128 | 256 | 512 | 1024 |
|---|---|---|---|---|---|
| LLaMA-1-7B | 29.58 | 31.72 | 33.97 | 41.29 | 146.46 |
| LLaMA-2-7B | 27.12 | 27.93 | 50.62 | 54.68 | 507.44 |

## 5 CONCLUSION

In this paper, we introduce STBLLM, a structured Binary LLM PTQ framework designed for sub-1-bit quantization. We address redundancy in binarized LLMs, highlighting the potential for further compression. Specifically, we present a Standardized Importance (SI) metric for N:M structured pruning. Then, we use the Hessian matrix to partition weights into salient and non-salient categories. We propose Non-salient Aware Quantization for non-salient weights, identifying optimal splitting points to create sparse, intermediate, and dense regions, each with tailored binarization. Finally, we design a specialized CUDA kernel with a sparse tensor core to achieve significant speedup. We validate the performance of STBLLM across LLaMA-1/2/3, OPT, and Mistral, demonstrating that STBLLM achieves a superior trade-off at sub-1-bit settings. By achieving LLM performance under 1 bit, STBLLM highlights the potential of extreme LLM compression. **Limitation:** STBLLM does not support Mixture of Experts (MoE) or Mamba-based language models.

ACKNOWLEDGMENT

This work was partially supported by National Natural Science Foundation of China under Grant No. 62272122, the Guangzhou Municipal Joint Funding Project with Universities and Enterprises under Grant No. 2024A03J0616, Guangzhou Municipality Big Data Intelligence Key Lab (2023A03J0012), and Theme-based Research Scheme (T45-205/21-N) from Hong Kong RGC, Generative AI Research and Development Centre from InnoHK, Hong Kong CRF grants under Grant No. C7004-22G and C6015-23G.

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

APPENDIX OVERVIEW

- Section A: STBLLM Implementation.
- Section B: Details of Motivation Experiments.
- Section C: Details of Hardware Accelerations.
- Section D: Impact of Extreme Weight on the Hessian Matrix.
- Section E: More Experimental Results and Ablation Study.

## A  STBLLM IMPLEMENTATION

Building on BiLLM (Huang et al., 2024), STBLLM preserves operations on salient weights while prioritizing non-salient weights. As detailed in Algorithm 2, the **BINARY** and **RES_APPROX** functions remain consistent with BiLLM, whereas we introduce the **NonSalientAwareQuant** and **Trisection** functions to enhance the framework.

For **NonSalientAwareQuant** function, it aims to find two optimal break-points to partition the symmetric Gaussian distribution of non-salient weight.

The naive implementation requires over 6 hours for LLaMA-2-7B, rendering it infeasible for 70B-scale LLMs. A naive approach for searching the break-point is using two nested loops, whose complexity is $O(N^2)$, where $N$ denotes the length of the search space. To reduce the complexity to $O(N)$, we propose to utilize $p_2 = \sigma \times p_1$ to locate the $p_2$. It is natural to assume that $p_2 > p_1$ and we have $\sigma > 1$.

To address this, we introduce a fixed ratio $\alpha$ to mitigate the computational overhead. Specifically, as illustrated in Figure 3(c), non-salient weights conform to a Gaussian distribution, $w \sim \mathcal{N}(\mu, \sigma^2)$, with the probability density function given by:

$$f(w) = \frac{1}{\sqrt{2\pi}\sigma} \exp\left(-\frac{w^2}{2\sigma^2}\right). \tag{7}$$

Leveraging the symmetry of the Gaussian distribution, we define the probabilities for three partitions:

- Sparse partition: $P_{\text{Sparse}} = 2\int_{p_2}^{\infty} f(w)\,dw = 2Q\left(\frac{p_2}{\sigma}\right)$
- Intermediate partition: $P_{\text{Intermediate}} = 2\int_{p_1}^{p_2} f(w)\,dw = 2\left[Q\left(\frac{p_1}{\sigma}\right) - Q\left(\frac{p_2}{\sigma}\right)\right]$
- Dense partition: $P_{\text{Dense}} = \int_{-p_1}^{p_1} f(w)\,dw = 1 - 2Q\left(\frac{p_1}{\sigma}\right)$,

where $Q(x) = \int_x^{\infty} \frac{1}{\sqrt{2\pi}} e^{-t^2/2} dt$ denotes the Gaussian tail probability. To ensure equal partition areas, we set $P_{\text{Sparse}} = P_{\text{Intermediate}} = P_{\text{Dense}} = \frac{1}{3}$, yielding:

$$Q\left(\frac{p_1}{\sigma}\right) = \frac{1}{3}, \quad Q\left(\frac{p_2}{\sigma}\right) = \frac{1}{6}, \quad Q\left(\frac{p_1}{\sigma}\right) - Q\left(\frac{p_2}{\sigma}\right) = \frac{1}{6}. \tag{8}$$

Solving these, we obtain $\frac{p_1}{\sigma} = Q^{-1}\left(\frac{1}{3}\right)$ and $\frac{p_2}{\sigma} = Q^{-1}\left(\frac{1}{6}\right)$. Using standard normal inverse $Q$-function values, we find $p_2 \approx 2p_1$, implying that the parameter $\sigma = 2$ in Algorithm 2.

For **Trisection** function, it aims to partition the symmetric Gaussian distribution presented in Figure 3(c) into three parts, which are Sparse, Intermediate, and Dense region. These three parts have no intersection and by uniting them together, we have all of the non-salient structured binarized weight.

Figure 5 provides a detailed illustration of our weight matrix partitioning strategy, complementing the overview presented in Figure 3(b). The figure demonstrates how we first partition weights into salient and non-salient regions based on Hessian matrix. For the salient weights, which constitute a small portion of the total weights, we employ residual approximation following BiLLM (Huang et al., 2024). The non-salient weights undergo our novel trisection partitioning scheme, where they are further divided into three distinct regions (dense, intermediate, and sparse) for optimized quantization. This hierarchical partitioning enables fine-grained control over compression while preserving model performance. The visualization shows how each region is processed differently, with

---

**Algorithm 2** STBLLM

---

func Salient ($\mathbf{W}, \mathbf{H^c}$)

1: **function** SALIENT($\mathbf{W}, \mathbf{H^c}$)
2:     $\mathbf{S} \leftarrow \mathbf{W}^2/[\mathbf{H}^c_{b:b+\beta;b:b+\beta}]^2$   ▷ Salient matrix
3:     $row_s \leftarrow \text{topk}(\text{sum}(\text{abs}(\mathbf{S})), \text{dim} = 0)$
4:     $e \leftarrow \infty$       ▷ Searching error
5:     $n^* \leftarrow 0$ ▷ Optimal number of salient columns
6:     **for** $i = 1$ to $\text{len}(row_s)$ **do**
7:         $\mathbf{B}_1 \leftarrow \text{binary}(\mathbf{W}_{:,j}, j \in row_s[: i])$
8:         $\mathbf{B}_2 \leftarrow \text{binary}(\mathbf{W}_{:,j}, j \notin row_s[: i])$
9:         **if** $\|\mathbf{W} - (\mathbf{B}_1 \cup \mathbf{B}_2)\|^2 < e$ **then**
10:           $e \leftarrow \|\mathbf{W} - (\mathbf{B}_1 \cup \mathbf{B}_2)\|^2$
11:           $n^* \leftarrow i$
12:         **end if**
13:     **end for**
14:     **return** $row_s[: n^*]$
15: **end function**

1: **function** BINARY($\mathbf{W}$)
2:     $\alpha \leftarrow \frac{\|\mathbf{W}\|_{\ell 1}}{m}$
3:     $\mathbf{B} \leftarrow \alpha \cdot \text{sign}(\mathbf{W})$
4:     **return** $\mathbf{B}$
5: **end function**

1: **function** RES_APPROX($\mathbf{W}$)
2:     $\mathbf{B}_1 \leftarrow \text{BINARY}(\mathbf{W})$
3:     $\mathbf{R} \leftarrow \mathbf{W} - \mathbf{B}_1$
4:     $\mathbf{B}_2 \leftarrow \text{BINARY}(\mathbf{R})$
5:     $\mathbf{B} \leftarrow \mathbf{B}_1 + \mathbf{B}_2$
6:     **return** $\mathbf{B}$
7: **end function**

1: **function** NONSALIENTAWAREQUANT($\mathbf{W}$)
2:     $e \leftarrow \infty$       ▷ Searching error
3:     $p_1^* \leftarrow 0$   ▷ Optimal break-point for trisection
4:     $p_2^* \leftarrow 0$   ▷ Optimal break-point for trisection
5:     **for** $i \in \text{np.linspace}(0.1, 0.9, 160)$ **do**
6:         $p_1 \leftarrow i \cdot \max(|\mathbf{W}|)$
7:         $p_2 \leftarrow \sigma \times p_1$
8:         **if** $p_2 > 0.9 \times \max(|\mathbf{W}|)$ **then**
9:           **continue**
10:         **end if**
11:         $\mathbf{B}_1 \leftarrow \text{BINARY}(\mathbf{W}_{|w_{i,j}|>p_2})$
12:         $\mathbf{B}_2 \leftarrow \text{BINARY}(\mathbf{W}_{p_1<|w_{i,j}|\leq p_2})$
13:         $\mathbf{B}_3 \leftarrow \text{BINARY}(\mathbf{W}_{|w_{i,j}|\leq p_1})$
14:         **if** $\|\mathbf{W} - (\mathbf{B}_1 + \mathbf{B}_2 + \mathbf{B}_3)\|^2 < e$ **then**
15:           $e \leftarrow \|\mathbf{W} - (\mathbf{B}_1 + \mathbf{B}_2 + \mathbf{B}_3)\|^2$
16:           $p_1^* \leftarrow p_1$
17:           $p_2^* \leftarrow p_2$
18:         **end if**
19:     **end for**
20:     **return** $p_1^*, p_2^*$
21: **end function**

1: **function** TRISECTION($\mathbf{W}, p_1^*, p_2^*$)
2:     $\tilde{\mathbf{B}}_2 \leftarrow \text{BINARY}(\mathbf{W}_{|w_{i,j}|>p_2^*})$
3:     $\tilde{\mathbf{B}}_3 \leftarrow \text{BINARY}(\mathbf{W}_{p_1^*<|w_{i,j}|\leq p_2^*})$
4:     $\tilde{\mathbf{B}}_4 \leftarrow \text{BINARY}(\mathbf{W}_{|w_{i,j}|\leq p_1^*})$
5:     **return** $\tilde{\mathbf{B}}_2, \tilde{\mathbf{B}}_3, \tilde{\mathbf{B}}_4$
6: **end function**

---

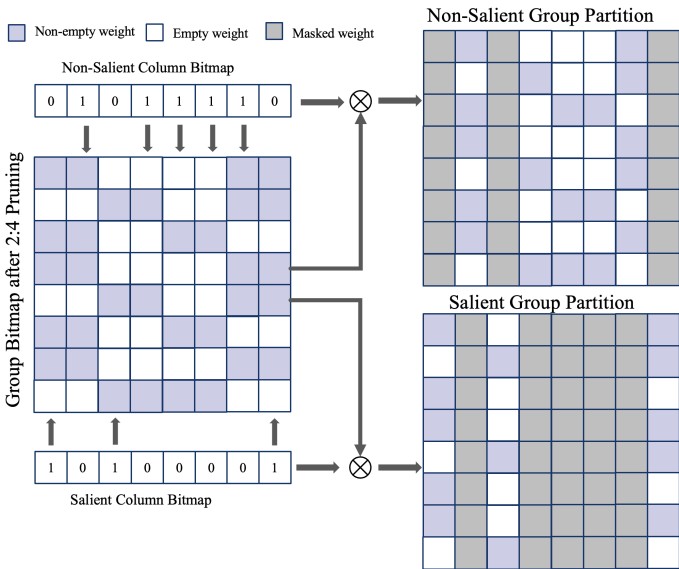

Figure 5: Column bitmap under 2:4 semi-structure pruned binarized matrix partition illustration.

the trisection boundaries clearly delineating the transitions between dense, intermediate, and sparse non-salient weight regions.

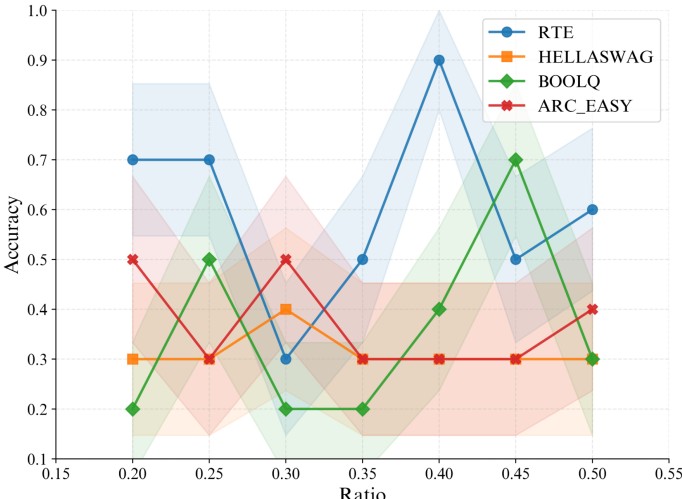

Figure 6: The impact of random flipping non-salient binarized weights on accuracy in 1-Bit LLaMA-2-7B with group size of 512.

## B  DETAILS OF MOTIVATION EXPERIMENT

In this section, we delineate the specifics of the motivation experiments as illustrated in Figure 1. Initially, we elucidate the procedure for inverting the signs of elements within a matrix, as detailed in Algorithm 3, to examine its effects on various computational tasks. This algorithm is designed to efficiently invert the signs of a specified proportion of elements in a given matrix $\mathbf{W}$. Subsequently, we employ this function on $RES\_APPROX$ and invert the signs of each binary matrix, including $B_1$ and $B_2$.

---

**Algorithm 3** Algorithm for Efficiently Flipping Signs of Matrix Elements.

1: **function** FLIPSIGNSEFFICIENT($\mathbf{W}$, ratio, $\mathbf{C} \leftarrow$ None)
2:     $n \leftarrow$ numel($\mathbf{W}$)                                    ▷ Total number of elements in $\mathbf{W}$
3:     $k \leftarrow$ int($n \times$ ratio)                              ▷ Number of elements to flip
4:     **if** $\mathbf{C} \neq$ None **then**
5:         **assert** shape($\mathbf{C}$) = shape($\mathbf{W}$)                 ▷ Ensure $\mathbf{C}$ matches $\mathbf{W}$
6:         _, idx $\leftarrow$ sort($\mathbf{C}$.view($-1$))            ▷ Flatten $\mathbf{C}$ and get sorted indices
7:         idx_to_flip $\leftarrow$ idx[: $k$]              ▷ Select least significant elements to flip
8:     **else**
9:         idx_to_flip $\leftarrow$ random_indices($0, n, k$)          ▷ Random select elements to flip
10:     **end if**
11:     $\mathbf{W_{flip}} \leftarrow \mathbf{W}$.clone()                                 ▷ Create a copy of $\mathbf{W}$
12:     $\mathbf{W_{flip-flat}} \leftarrow \mathbf{W_{flip}}$.view($-1$)            ▷ View the copy as a 1D tensor
13:     $\mathbf{W_{flip-flat}}$[idx_to_flip] $\leftarrow \mathbf{W_{flip-flat}}$[idx_to_flip] $\times -1$          ▷ Flip the signs of selected elements
14:     **return** $\mathbf{W_{flip}}$
15: **end function**

---

We further visualize the impact of random flipping non-salient binarized weights on accuracy at higher ratios (from 0.2 to 0.5) in Figure 6. As shown in the table below, we find that as the ratio increases, the performance fluctuates but does not deteriorate drastically across multiple tasks like RTE (accuracy ranging from 0.3-0.9), HellaSwag (stable around 0.3), BoolQ (0.2-0.7), and ARC Easy (0.3-0.5). This suggests a degree of robustness in our approach to varying ratios of flipped non-salient weights.

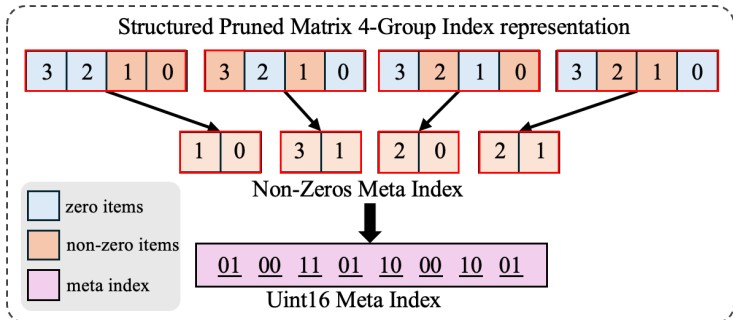

Figure 7: Structured Pruned Matrix 4-Group Index Representation for 2:4 Structured Sparsity Acceleration.

## C  DETAILS OF HARDWARE ACCELERATION

### C.1  IMPLEMENTATION DETAILS

Recent advancements in low-precision computing have significantly enhanced the practical implementation of efficient neural network techniques. A prime example is the introduction of Ladder (Wang et al., 2024), released as BitBLAS, a software library that seamlessly integrates into existing Deep Neural Network (DNN) and Large Language Model (LLM) frameworks. This integration enables highly efficient low-precision computations across various hardware platforms. The impact of these developments is evident in popular frameworks like llama.cpp (Gerganov, 2023), which now supports 1.5-bit quantization through BitNet (Wang et al., 2023a). This advancement has resulted in impressive performance gains, achieving 198 tokens per second on a single CPU core. Moreover, for large-scale models such as LLaMA-2-70B, the implementation of Ladder (Wang et al., 2024) to accelerate BitNet 1.58 (Wang et al., 2023a) has yielded remarkable results, demonstrating a $4.6\times$ speedup compared to FP16 precision.

The emergence of Sparse Tensor Cores (SPTCs) since NVIDIA's Ampere architecture has revolutionized the processing of sparse matrices, offering an efficient mechanism for handling 50% sparsity. Theoretically, by eliminating half of the computations, SPTCs can potentially double the computational power compared to Dense Tensor Cores. There are already several research over accelerating dense tensor core, including ULPPack (Won et al., 2022), NGEMM (Bao et al., 2019) and QQQ (Zhang et al., 2024a). However, efficiently representing 1-bit values ($+1$, $-1$, and 0 for sparsity) and achieving sufficient real-world acceleration pose significant challenges. To address these, we propose a novel 2-bit integer representation method, particularly useful for the General Matrix Multiply (GEMM) operation, formulated as $D = A(E) \times B + C$. Here, $A$ represents the 2:4 1-bit sparse matrix, $B$, $C$, and $D$ are dense tensors, and $E$ employs `uint16` to denote the valid indices of $A$.

Our approach introduces a 6-bit encoding scheme for each group of 2:4 sparse 1-bit values. This scheme comprises four bits for indexing and two bits for physical value representation, where $1 \rightarrow +1$, $0 \rightarrow -1$, and positions unmarked by $E$ indicate sparsity (0). This method significantly improves memory efficiency compared to a baseline approach using 2-bit integers to represent $-1$, 0, and $+1$, which would require 8 bits for an equivalent group size. Consequently, our encoding method reduces memory footprint by approximately 25%, leading to decreased global memory access requirements. In memory-bound scenarios typical of large-scale model inference, this approach theoretically offers up to a 1.333-fold increase in processing speed compared to the 2-bit variant.

To fully leverage these optimizations, we employ semi-structured pruning techniques specifically tailored for NVIDIA's GPU architecture. These techniques enable the use of Sparse Tensor Cores optimized for processing sparse matrices. By structuring the sparsity (e.g., $N:M$ sparsity where $N$ out of $M$ weights are non-zero), we can effectively utilize the Sparse Tensor Cores, leading to substantial improvements in processing speed and efficiency. Specifically, the process of implementing these optimizations involves several key steps in matrix compression and manipulation:

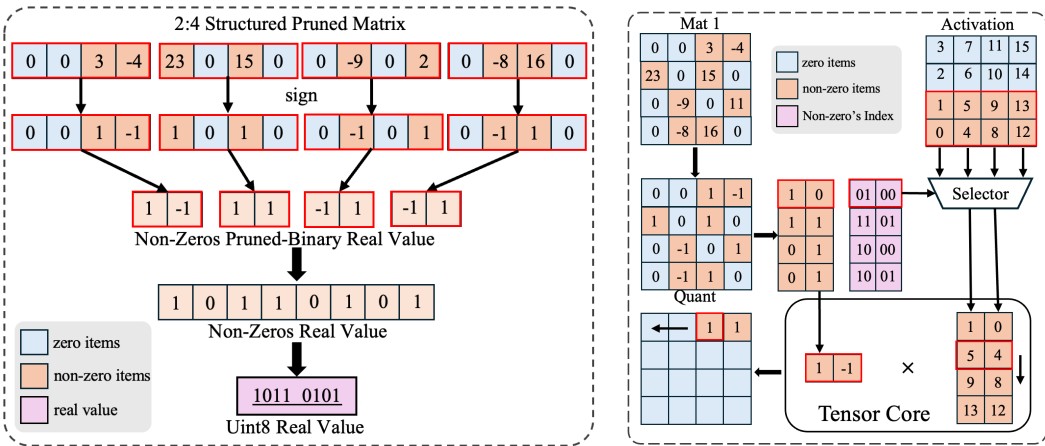

Figure 8: The overview of sparsity pattern of 1-bit kernel that convert weight matrix to structured pruned matrix.

Figure 9: 2:4 Structured Sparsity Matrix Multiplication Using Tensor Cores.

1. **Matrix Compression:** The input matrix is partitioned into 4-element groups as shown in Figure 7. Zero elements are identified and non-zero elements are extracted. The positions of non-zero elements are recorded in a Non-Zeros Meta Index, which is then encoded into a compact `Uint16` Meta Index. This encoding facilitates efficient localization of non-zero elements during matrix operations, enhancing computational speed by enabling the omission of zero elements.

2. **Value Compression:** Similar to matrix compression, the matrix is divided into 4-element groups as shown in Figure 8. The sign of each non-zero element is extracted to form the Non-Zeros Pruned-Binary Real Value. These values are then converted to a binary format, creating the Non-Zeros Real Value. Finally, the each of eight binary values are concatenated into a compact `Uint8` Real Value, optimizing storage and computation by focusing on the non-zero elements and their signs.

3. **Matrix Multiplication with Structured Sparsity:** The input matrices undergo pruning to retain only non-zero elements and their corresponding indices as shown in Figure 9. The pruned matrices are then quantized, extracting non-zero values and their positions. Both processed matrices are subsequently input into the Sparse Tensor Cores, which executes efficient multiplication by focusing on the non-zero elements, resulting in a compressed and accelerated computation.

## C.2 THEORY ANALYSIS

To evaluate the performance of various matrix multiplication algorithms across different problem sizes, we present a comprehensive roofline model analysis in Figure 10. Each subplot depicts the relationship between arithmetic intensity (FLOPS/Byte) and performance (TFLOPS). During prefilling stage, $N$ denotes the product of sequence length and batch size. For decoding stage, $N$ denotes the batch size. $M$ and $K$ correspond to the dimension of weight matrix. To compare different implementations, we include standard FP16 GEMM, 2-bit quantized GEMM, and 1-bit 2vs4 quantized GEMM, alongside theoretical performance limits represented by roofline models for Tensor Core and Tensor Core Sparse operations. From our analysis, we observe that as N increases, all algorithms exhibit improved performance, with quantized versions consistently outperforming standard GEMM. We find that our 1-bit 2vs4 quantized GEMM demonstrates superior performance, particularly at larger N values, often approaching the Tensor Core Sparse roofline.

The advantages of our 1-bit 2:4 quantized GEMM kernel arise from reduced memory access overhead and the higher compute upper bound of Sparse Tensor Cores (SPTCs). When $N$ is small (particularly during the decoding phase), all GEMM kernels are memory-bound, but our 1-bit 2:4 quantized GEMM kernel achieves relatively better performance due to its higher compression rate. As $N$ increases (especially during the prefilling stage), the quantized GEMM kernels tend to become

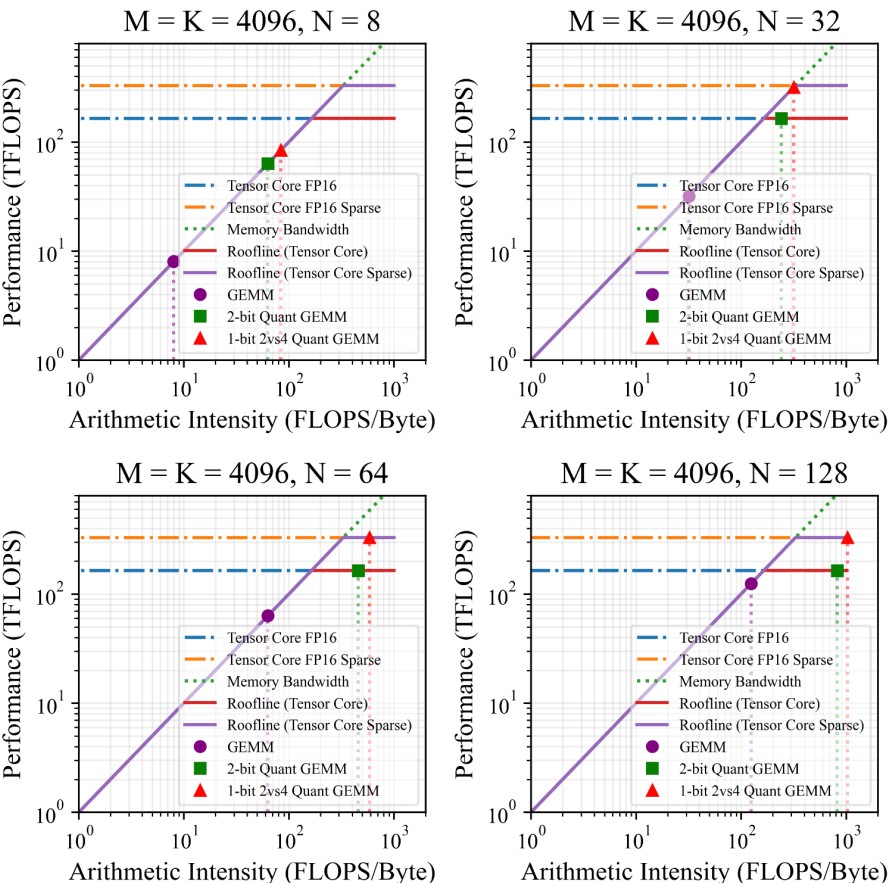

Figure 10: Roofline for Sparse GEMM Quantization.

compute-bound. In this case, our specialized GEMM kernel can theoretically achieve a $2\times$ speedup compared to other GEMM kernels.

This extreme quantization approach significantly reduces both computational overhead and memory footprint by limiting the precision of weights to just two possible states. Such a method is particularly advantageous in resource-constrained environments, improving the deployability of large models on devices with limited hardware capabilities.

Given the success of the 1-bit 2:4 quantized GEMM kernel in improving the efficiency of large language models by reducing memory and computational demands, we now plan to extend the STBLLM framework to compress Mixture of Experts (MoE) (Gu et al., 2025; Li et al., 2025). By leveraging AutoML (Li et al., 2024a; Dong et al., 2023a; Li et al., 2023; 2024c), we aim to automate and optimize the compression process for these complex models, ensuring they can also be efficiently deployed on resource-constrained devices.

## C.3 MEMORY COMPARISON

As illustrated in Figure 11, we present the memory consumption of FP16, CUTLASS, ABQ-LLM, and our implementation for LLaMA-7B, 13B, and 30B models. Our proposed methodology demonstrates a substantial memory compression gain, exceeding 3.1 times that of SmoothQuant. This performance significantly surpasses current mainstream inference techniques. Furthermore, our approach achieves an approximate 15% reduction in memory usage compared to ABQ-LLM. These notable improvements have important implications for the field of large language models (LLMs). By reducing the memory footprint, our method decreases the operational costs associated with LLM services and facilitates their practical deployment in real-world applications.

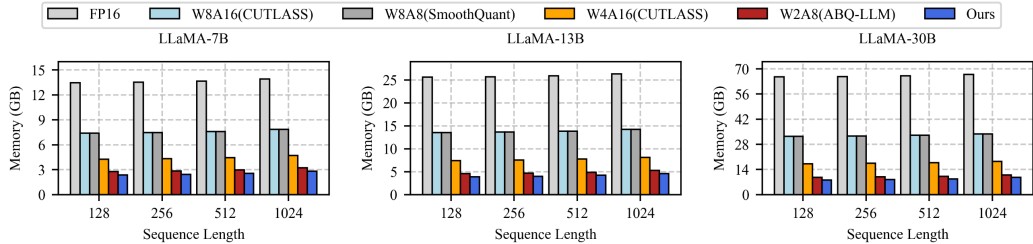

Figure 11: Memory Usage Comparison of Various Quantization Methods for LLaMA Models.

Table 10: Comparison of Quant-Only, Structure-Only, and ours across different datasets.

| | LLaMA-1-7B | | | LLaMA-2-7B | | |
|---|---|---|---|---|---|---|
| Dataset | Quant-Only | Structure-Only | Ours | Quant-Only | Structure-Only | Ours |
| PTB | 23.52 | 14.24 | 68.48 | 2071.44 | 69.25 | 690.76 |
| C4 | 15.75 | 10.52 | 36.04 | 14.62 | 10.29 | 30.81 |
| Wikitext2 | 12.29 | 8.13 | 31.72 | 11.17 | 7.85 | 27.93 |

## D  IMPACT OF EXTREME WEIGHT ON THE HESSIAN MATRIX

The Hessian matrix $H$ is defined as: $H_{ij} = \frac{\partial^2 L}{\partial w_i \partial w_j}$, where $L$ is the loss function, and $w_i$ and $w_j$ are weights. If a weight $w_k$ has extreme values, the corresponding elements in the Hessian matrix, particularly $H_{kk}$, will be significantly larger than others.

For instance, if $w_1$ is an extreme value, the Hessian matrix might look like:

$$H = \begin{pmatrix} h_{11} & h_{12} & \cdots & h_{1n} \\ h_{21} & h_{22} & \cdots & h_{2n} \\ \vdots & \vdots & \ddots & \vdots \\ h_{n1} & h_{n2} & \cdots & h_{nn} \end{pmatrix}$$

Here, $h_{11}$ is much larger than other elements. This disproportionate value significantly influences the Hessian's eigenvalues, with at least one eigenvalue becoming very large. During optimization, methods like Newton's method update weights using the inverse of the Hessian matrix:

$$\mathbf{w}_{\text{new}} = \mathbf{w} - \eta H^{-1} \nabla L(\mathbf{w}),$$

where $\eta$ is the learning rate, and $\nabla L(\mathbf{w})$ is the gradient. The presence of an extreme value in $h_{11}$ causes the corresponding element in $H^{-1}$ to be very small, affecting the step size in weight updates:

$$\Delta w_1 \approx -\eta \frac{\partial L}{\partial w_1} / h_{11},$$

$$\Delta w_2 \approx -\eta \frac{\partial L}{\partial w_2} / h_{22}.$$

Since $h_{11}$ is large, $\Delta w_1$ becomes small, indicating minimal adjustments for the extreme value weight, while $\Delta w_2$ remains relatively larger for the normal weights.

## E  MORE EXPERIMENTAL RESULTS

### E.1  MODULE ABLATION STUDY

To evaluate the interdependent interaction between quantization and pruning within our STBLLM framework, we conduct a module ablation study. This study isolates the effects of quantization-only, pruning-only, and our combined method on the performance of the LLaMA-1-7B and LLaMA-2-7B models across the PTB, C4, and Wikitext2 datasets. The results are presented in Table 10.

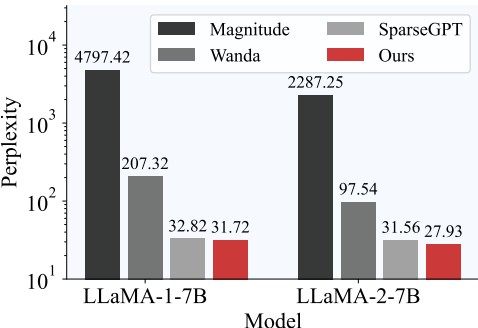 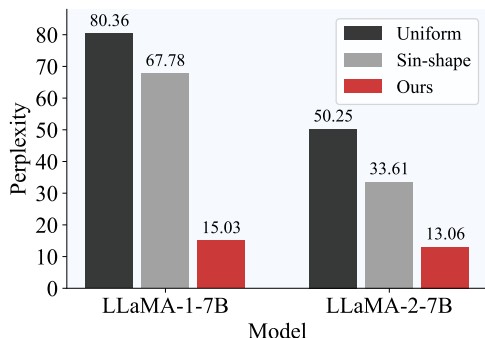

Figure 12: **Ablation study on post-training pruning metrics on STBLLM on LLaMA-1-7B and LLaMA-2-7B.** Our method achieves the best performance among these metrics.

Figure 13: **Ablation study on allocation strategies on STBLLM on LLaMA-1-7B and LLaMA-2-7B.** Our method achieves nearly identical perplexity.

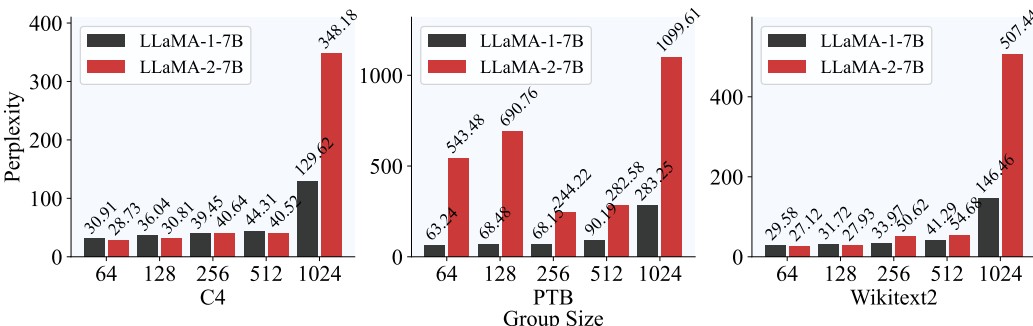

Figure 14: **Comparison across different sizes for LLaMA-1-7B and LLaMA-2-7B.**

The ablation results highlight the synergistic effect of combining quantization and pruning in our approach, significantly outperforming each method applied in isolation.

**LLaMA-1-7B Analysis**

- *PTB Dataset*: Our combined method achieves a score of 68.48, markedly higher than quantization-only (23.52) and pruning-only (14.24). This demonstrates the substantial performance gains achieved by leveraging the complementary strengths of both techniques.

- *C4 Dataset*: Our method scores 36.04, compared to 15.75 for quantization-only and 10.52 for pruning-only. The combined approach effectively mitigates the limitations of individual methods, resulting in superior performance.

- *Wikitext2 Dataset*: The score of 31.72 for our method far exceeds the results of quantization-only (12.29) and pruning-only (8.13), underscoring the enhanced model efficiency and accuracy through our integrated approach.

**LLaMA-2-7B Analysis**

- *PTB Dataset*: Although quantization-only achieves an unusually high score of 2071.44, our combined method still significantly outperforms pruning-only (690.76 vs. 69.25). This suggests that while quantization might retain certain advantageous structures, the integration with pruning leads to a more balanced and robust model.

- *C4 Dataset*: The combined method's score of 30.81 surpasses quantization-only (14.62) and pruning-only (10.29), highlighting the effectiveness of our method in maintaining high performance across varying model versions.

Table 11: Comparison of C4, PTB, and Wikitext2 across LLaMA-1-7B and LLaMA-2-7B

| | LLaMA-1-7B | | | LLaMA-2-7B | | |
|---|---|---|---|---|---|---|
| Dataset | C4 | PTB | Wikitext2 | C4 | PTB | Wikitext2 |
| C4 | 36.04 | 68.48 | 31.72 | 30.81 | 690.76 | 27.93 |
| PTB | 54.57 | 35.13 | 49.27 | 43.04 | 4569.03 | 40.94 |
| Wikitext2 | 40.76 | 71.81 | 20.48 | 37.01 | 1970.76 | 20.60 |

Table 12: Comparison across different sizes for LLaMA-1-7B and LLaMA-2-7B

| | LLaMA-1-7B | | | LLaMA-2-7B | | |
|---|---|---|---|---|---|---|
| Group Size | C4 | PTB | Wikitext2 | C4 | PTB | Wikitext2 |
| 64 | 30.91 | 63.24 | 29.58 | 28.73 | 543.48 | 27.12 |
| 128 | 36.04 | 68.48 | 31.72 | 30.81 | 690.76 | 27.93 |
| 256 | 39.45 | 68.15 | 33.97 | 40.64 | 244.22 | 50.62 |
| 512 | 44.31 | 90.19 | 41.29 | 40.52 | 282.58 | 54.68 |
| 1024 | 129.62 | 283.25 | 146.46 | 348.18 | 1099.61 | 507.44 |

- *Wikitext2 Dataset*: Our method's score of 27.93 is higher than both quantization-only (11.17) and pruning-only (7.85), further confirming the synergistic benefits of combining these techniques.

### E.2 ABLATION STUDY OF CALIBRATION DATASET

Table 11 presents an ablation study comparing the performance of LLaMA-1-7B and LLaMA-2-7B models when trained on different calibration datasets: C4, PTB, and Wikitext2. The purpose of this experiment is to investigate how the choice of calibration dataset affects the models' performance on various evaluation datasets.

In this study, both LLaMA-1-7B and LLaMA-2-7B models are trained on each of the three calibration datasets separately. The trained models are then evaluated on all three datasets, resulting in a 3x3 matrix of performance scores for each model.

The performance scores in the table likely represent some evaluation metric, such as perplexity or loss, where lower values indicate better performance. The diagonal values (e.g., C4 evaluated on C4) represent in-domain performance, while off-diagonal values represent out-of-domain performance.

### E.3 ABLATION STUDY OF GROUP SIZE

Table 12 and Figure 14 presents an ablation study that compares the performance of LLaMA-1-7B and LLaMA-2-7B models across different group sizes. The purpose of this experiment is to investigate how the choice of group size affects the models' performance on various evaluation datasets.

In this study, both LLaMA-1-7B and LLaMA-2-7B models are trained with different group sizes: 64, 128, 256, 512, and 1024. The trained models are then evaluated on three datasets: C4, PTB, and Wikitext2. The performance scores in the table likely represent some evaluation metric, such as perplexity or loss, where lower values indicate better performance. By comparing the performance scores across different group sizes and evaluation datasets, researchers can gain insights into the impact of group size on the models' performance and generalization capabilities.

The results show that the performance of both models varies with the choice of group size. For LLaMA-1-7B, the best performance on C4 and Wikitext2 is achieved with a group size of 64, while for PTB, the best performance is obtained with a group size of 128. For LLaMA-2-7B, the best performance on C4 and Wikitext2 is also achieved with a group size of 64, while for PTB, the best performance is obtained with a group size of 256. Interestingly, the performance of both models deteriorates significantly when the group size is increased to 1024, suggesting that excessively large group sizes may lead to overfitting or other training issues.

Table 13: Motivation: The perplexity on different top percentages.

| Top Percentage | Perplexity | Top Percentage | Perplexity | Top Percentage | Perplexity |
|---|---|---|---|---|---|
| 0.01 | 27.770422 | 0.02 | 30.168285 | 0.03 | 34.049734 |
| 0.04 | 36.191769 | 0.05 | 33.821476 | 0.06 | 36.452296 |
| 0.07 | 38.702617 | 0.08 | 39.169894 | 0.09 | 44.818825 |
| 0.10 | 54.451229 | 0.11 | 49.835159 | 0.12 | 71.762848 |
| 0.13 | 52.129317 | 0.14 | 52.568348 | 0.15 | 65.945448 |
| 0.16 | 62.712751 | 0.17 | 117.990227 | 0.18 | 138.912356 |

The provided Figure 1 and Table 13 present an experiment that investigates the relationship between the top Percentage of data and the corresponding perplexity scores in a LM. The purpose of this experiment is to understand how the choice of top Percentage affects the model's performance and to determine an optimal threshold for data selection. In this experiment, we randomly flip 1%-16% weights from binarized LM and evaluate their downstream tasks' performance including ARC (Clark et al., 2018), BoolQ (Mihaylov et al., 2018), Hellaswag (Zellers et al., 2019) and RTE (Chakrabarty et al., 2021). Table 13 shows the perplexity scores for each top Percentage. Lower perplexity scores indicate better language model performance, as the model is better able to predict the next word in a sequence.

Figure 1 provides a visual representation of the relationship between the top Percentage and perplexity scores. It shows that the perplexity scores initially improve as the top Percentage increases, indicating that including more high-quality data points benefits the model's performance. However, beyond a certain threshold (around 0.05 to 0.10), the perplexity scores start to deteriorate, suggesting that including lower-quality data points negatively impacts the model's performance.

### E.4 ABLATION STUDY: PRUNE-THEN-QUANTIZE VS. QUANTIZE-THEN-PRUNE

To validate our choice of prune-then-quantize strategy, we conduct an ablation study comparing it with the alternative quantize-then-prune approach. As shown in Table 14, the prune-then-quantize approach consistently achieves better perplexity scores across both LLaMA-1-7B and LLaMA-2-7B models.

Table 14: Comparison of Pruning and Quantization Order (6:8 ratio).

| Approach | Averaged Bits | Model | Perplexity |
|---|---|---|---|
| Prune $\rightarrow$ Quantize | 0.82 | LLaMA-1-7B | 15.03 |
| Prune $\rightarrow$ Quantize | 0.80 | LLaMA-2-7B | 13.06 |
| Quantize $\rightarrow$ Prune | 0.82 | LLaMA-1-7B | 34.02 |
| Quantize $\rightarrow$ Prune | 0.80 | LLaMA-2-7B | 31.98 |

These results empirically support our design choice, showing that prune-then-quantize achieves significantly lower perplexity compared to quantize-then-prune. Our analysis suggests this is because quantization typically causes less performance degradation compared to pruning. When applying the more damaging operation (pruning) after quantization, it becomes more challenging to recover performance through block-wise OBC. Conversely, applying quantization after pruning allows for better performance recovery.

### E.5 ABLATION STUDY: IMPACT OF WEIGHT PARTITIONING STRATEGIES

To investigate the effectiveness of different weight partitioning approaches, we conducted an ablation study comparing various partitioning strategies under identical experimental conditions using the LLaMA-2-7B model with 6:8 sparsity ratio:

The bell-shaped distribution approach, originally proposed in BiLLM, and our non-salient partitioning strategy in STBLLM demonstrate comparable computational efficiency with search times of approximately 0.5 hours. While the naive implementation method achieves a slightly lower perplex-

Table 15: Comparison of Different Weight Partitioning Strategies and their Search Time.

| # Partitions | Perplexity | Search Time |
|---|---|---|
| 1 (Bell-shaped) | 50.25 | ∼0.5h |
| 2 (Non-salient) | 13.06 | ∼0.5h |
| 2 (Naive implementation) | 12.78 | ∼6h |

ity score of 12.78 compared to our non-salient approach (13.06), it requires a significantly longer search time of approximately 6 hours - a 12-fold increase in computational cost.

Based on these empirical results, we adopt T=2 (three partitions) for our non-salient partitioning strategy as it provides an optimal balance between granular weight importance differentiation and computational efficiency.

### E.6    ABLATION STUDY: IMPACT OF DIFFERENT PRUNING METHODS

To comprehensively evaluate the effectiveness of different pruning methods, we conducted experiments using both Wanda and SI pruning on different weight distributions in LLaMA-1-7B and LLaMA-2-7B models. Table 16 presents the percentage of weights affected by each pruning method across different weight distributions.

Table 16: Impact of Different Pruning Methods on Weight Distributions.

| Model | Distribution | Method | Percentage (%) |
|---|---|---|---|
| LLaMA-1-7B | Bell-shaped | Wanda | 80.35 |
| | Non-salient | SI | 15.03 |
| | Bell-shaped | SI | 40.25 |
| | Non-salient | Wanda | 31.72 |
| LLaMA-2-7B | Bell-shaped | Wanda | 50.25 |
| | Non-salient | SI | 13.06 |
| | Bell-shaped | SI | 24.54 |
| | Non-salient | Wanda | 27.93 |

The results demonstrate that different pruning methods exhibit varying effectiveness depending on the weight distribution. For bell-shaped distributions, Wanda pruning affects a larger percentage of weights (80.35% and 50.25% for LLaMA-1-7B and LLaMA-2-7B respectively), while SI pruning shows better efficiency on non-salient weights (15.03% and 13.06%). This analysis supports our strategy of applying different pruning methods based on the weight distribution characteristics.

