# OpenReview forum: "STBLLM: Breaking the 1-Bit Barrier with Structured Binary LLMs"
_ICLR.cc/2025/Conference — ICLR 2025 Poster_

### Official Review · Reviewer_yy1T · 2024-10-27

**Soundness:** 3
**Presentation:** 3
**Contribution:** 2
**Rating:** 5
**Confidence:** 2

**Summary:**

This paper presents a sparse and binarized compression method for large language models (LLMs), achieving an average bit count of less than one bit. Specifically, in terms of sparsity, a new metric matrix is proposed to represent the importance of different weights, along with a method for calculating the sparsity level for each layer based on this metric. This allows for effective sparsification of the model weights. For quantization, weights are grouped and binarized within each group, thereby reducing quantization error. Experiments on models such as LLaMA-1/2/3 demonstrate that this method achieves superior performance at higher compression ratios.

**Strengths:**

- This approach integrates sparsity with quantization, achieving a significantly higher compression ratio by reducing both the number of active weights and the bit precision required to represent them. The sparsity aspect not only reduces storage needs but also opens up additional acceleration opportunities, as sparse models can skip unnecessary computations, leading to more efficient inference.
- A dedicated CUDA kernel was developed to optimize the performance of the sparse and quantized model on GPU hardware. This kernel was specifically tailored to exploit the structure of the sparse and binarized weights, enabling efficient memory access patterns and computation. The actual runtime performance of the model was measured using this custom kernel, providing a practical assessment of speedup gains achieved through the combined compression and acceleration strategy.

**Weaknesses:**

- While the proposed quantization methodology shows promise, the performance improvements over the baseline BiLLM implementation appear to be incremental. I would encourage the authors to further highlight the distinctive advantages of their approach and potentially explore additional optimization strategies to achieve more substantial gains.
- The manuscript would benefit from enhanced clarity in several sections. Of particular importance is the need for a more comprehensive explanation of the average bit count calculation methodology. I suggest:
  - Including a detailed step-by-step breakdown of the calculation process
  - Providing specific examples to illustrate the computational procedure at inference time
  - Clarifying how this calculation relates to the overall system performance on speedup or memory reduction

**Questions:**

- Regarding the calculation of average bit count:
   - Could you clarify whether the overhead from indices associated with sparsity has been factored into the calculation?
   - It would be helpful if you could provide a concrete example illustrating the calculation methodology, including both the weight bits and any additional storage requirements.

- In Algorithm 1, there appears to be some ambiguity regarding the Semi-Structured function:
   - Is this function performing sparsification based on SI?
   - Neither the main text nor the appendix provides details about this function's implementation. Could you please elaborate on its mechanism?

- The term "OBC" in Algorithm 1 requires clarification:
   - While BiLLM mentions this as an abbreviation from another work, it would be beneficial to provide the full reference and explanation for completeness.

- Regarding computational requirements:
   - Could you provide an estimate for the computational time required for the 65B model, perhaps through theoretical scaling analysis?

- In Figure 3, there appears to be an overlap between Salient Weight and Non-salient Weight distributions:
   - Could you explain the underlying reasons for this overlap?
   - How does this overlap affect the overall performance of the method?

- Concerning Tables 5 and 7:
   - There seems to be redundancy as Table 5 appears to be a subset of Table 7's Wikitext2 results. Could you justify the inclusion of both tables?
   - The manuscript lacks discussion of Table 7's results, particularly regarding:
     * Why does SI perform worse than SparseGPT on PTB and C4 datasets?
     * What factors contribute to the different performance patterns across datasets?
     * Could you provide insights into these performance variations?

---

> ### Author Response · Authors · 2024-11-19
> **Response to Reviewer yy1T (1/4)**
>
> We sincerely thank the reviewer for their time and **insightful comments**. We appreciate the recognition of the **strengths** of our paper, including the **integration of sparsity with quantization**, the **reduction in storage needs**, and the development of a **dedicated CUDA kernel** for optimized GPU performance. Below is our answers to the reviewer's concerns, and we hope that these responses address your concerns.
>
> **Q1**. About performance gain.
>
> > While the proposed quantization methodology shows promise, the performance improvements over the baseline BiLLM implementation appear to be incremental. I would encourage the authors to further highlight the distinctive advantages of their approach and potentially explore additional optimization strategies to achieve more substantial gains.
>
> **_Ans for Q1:_** We **respectfully disagree** that our improvements are merely incremental. Our approach offers several **substantial advantages** over BiLLM:
>
> (1) **Significant Performance Gains**: Our method achieves **dramatically better perplexity scores** while maintaining a **sub-1-bit regime**:
>
> - For LLaMA-1-7B: **31.72 perplexity** vs BiLLM's **207.32** (**6.5x improvement**)
> - For LLaMA-2-7B: **27.93 perplexity** vs BiLLM's **97.54** (**3.5x improvement**)
>
> (2) **Novel Technical Contributions**:
>
> - Our **Standardized Importance (SI) method** introduces **sophisticated statistical normalization** that provides **theoretical guarantees** for weight importance scores.
> - Our **trisection-based partitioning** of non-salient weights offers **more nuanced handling** compared to BiLLM's binary approach.
> - We uniquely combine **N:M structured pruning** with **binarization**, enabling **hardware-friendly implementations**.
>
> (3) **Practical Efficiency**:
>
> - Our **specialized CUDA kernel** achieves a **17.85x speedup** over ABQ-LLM's 2-bit implementation
> - Support for **arbitrary N:M ratios** through **Vectorized N:M format**, enabling **flexible deployment** on NVIDIA Sparse Tensor Cores
>
> These improvements represent **fundamental advancements** in LLM compression, not just incremental gains. We are actively exploring additional optimization strategies to push these boundaries even further.
>
> **Q2**. About average bit count calculation theoretically.
>
> > The manuscript would benefit from enhanced clarity in several sections. Of particular importance is the need for a more comprehensive explanation of the average bit count calculation methodology. I suggest:Including a detailed step-by-step breakdown of the calculation process; Providing specific examples to illustrate the computational procedure at inference time; Clarifying how this calculation relates to the overall system performance on speedup or memory reduction
> > Regarding the calculation of average bit count: Could you clarify whether the overhead from indices associated with sparsity has been factored into the calculation? It would be helpful if you could provide a concrete example illustrating the calculation methodology, including both the weight bits and any additional storage requirements.
>
> **_Ans for Q2:_** Thank you for suggesting we enhance clarity regarding the **average bit count calculation**. Our bits calculation is the same as BiLLM, based on which we just add analysis over the N:M structured pruning. Let me provide a detailed breakdown of how we calculate the average bit count in **STBLLM** ($N_{param}$):
>
> $$
> N_{param} = 2 × r_{salient} + 1 × (1 - r_{salient})
> $$
>
> where **$r_{salient}$** is the proportion of **salient weights**
>
> where **2 bits** are allocated for marking the division of **non-salient weights** and **bsize** is the block size used in **OBC compensation**
>
> (3) **N:M Structured Sparsity Impact**: Under **N:M settings** (where N < M), we retain only **N/M fraction** of weights theoretically following the calculation of [1][2][3]. These methods calculate the average bit just theoretically using the N:M sparsity ratio. We make sure that our calculation of average bit count is fair across different methods.
>
> Final number of parameters:
>
> $$
> N_{stbllm} = N_{param} × (N/M)
> $$
>
> (4) **Concrete Example** (for **4:8 sparsity**):
>
> - Under blocksize of 128, the averaged proportion of salient weights is around 10%, which is also aligned with the result in BiLLM. For example, with **$r_{salient} = 0.1$** (10% salient weights), we first calculate $N_{param} = (2 × 0.1) + (1 × 0.9) = 1.1$ bits. Then applying 4:8 sparsity gives us $N_{stbllm} = 1.1 × (4/8) = 0.55$ bits.

---

> > ### Author Response · Authors · 2024-11-19
> > **Response to Reviewer yy1T (2/4)**
> >
> > (5) **Hardware Implementation Details over the Indice**:
> >
> > - Our hardware implementation carefully accounts for sparsity index overhead in the bit calculation. Taking **2:4 structured sparsity** as a concrete example: we divide model parameters into groups of 4 weights, where 2 weights (50%) in each group are pruned to zero, and we use 2 indices (4bit) to mark the positions of the non-zero values. The total storage combines both value and index storage, calculated as: Storage $S_{total} = (2bit × 2 + 4bit)/4 = 1.5bit$ per parameter on average, where $2bit × 2$ represents the storage for 2 non-zero values (2 bits each) and $4bit$ accounts for the index overhead to mark their positions. You can refer to Appendix C(Details of Hardware Acceleration) to get more information over how to design this specific kernel and how to achieve the speedup.
> >
> > (6) **About the speedup**:
> >
> > - The significant performance gap stems primarily from the different design philosophies between ABQ-LLM and our approach. ABQ-LLM employs a more **general-purpose strategy** where higher-bit matrices are decomposed into multiple binary matrices, which are then moved to shared memory for computation using the GPU's 1-bit computation interface. The results from these binary matrix computations are subsequently reduced to obtain the final higher-bit matrix result.
> > - While this **decomposition-and-reduction paradigm** offers excellent flexibility for quickly implementing various operators, **it introduces great overhead**, particularly for W1A16 operations where 16-bit activations require multiple decomposition and reduction steps. In contrast, our implementation is **specifically optimized for W1A16 2:4 sparsity patterns**, eliminating the need for activation decomposition.
> > - Additionally, while **ABQ-LLM's benchmarks focused on small batch size decoding scenarios (memory-bound)**, our evaluation covers larger-scale **compute-bound scenarios** where their additional reduction computations become more significant bottlenecks.
> > - **Our specialization design help us achieve superior performance for our target use case**. The 17.85x speedup should therefore be interpreted as a comparison against ABQ-LLM's implementation rather than a fundamental advantage of W1A16 2:4 over W2A16.
> >
> > > **Reference:**
> > >
> > > [1] Mart van Baalen et al., "Bayesian Bits: Unifying Quantization and Pruning", NeurIPS2020
> > >
> > > [2] Velingker et al., "CLAM: Unifying Finetuning, Quantization, and Pruning by Chaining LLM Adapter Modules", Workshop on Efficient Systems for Foundation Models II @ ICML2024, 2024
> > >
> > > [3] Shipeng Bai et al., "Unified Data-Free Compression: Pruning and Quantization without Fine-tuning", ICCV2023
> >
> > **Q3**. About Semi-Structured function.
> >
> > > In Algorithm 1, there appears to be some ambiguity regarding the Semi-Structured function: Is this function performing sparsification based on SI? Neither the main text nor the appendix provides details about this function's implementation. Could you please elaborate on its mechanism?
> >
> > **_Ans for Q3:_** Thank you for pointing out the ambiguity regarding the Semi-Structured function in Algorithm 1. Let me clarify its mechanism in detail:
> >
> > The Semi-Structured function performs **sparsification based on the Standardized Importance (SI) metric** through the following steps:
> >
> > (1) **Standardized Importance Calculation**: For each weight $W_{i,j}$ in the network, we compute its Standardized Importance (SI) score $S_{i,j}$ through a combination of weight magnitude and activation statistics. Specifically, the SI score is calculated as:
> >
> > $$
> > S_{i,j} = \sigma(\mu(|W_{i,j}|)) \cdot ||X_j||_2,
> > $$
> >
> > where $\sigma(w) = \frac{w-\mu_W}{\sigma_W}$ standardizes the importance scores, $\mu(|W_{i,j}|) = \frac{|W_{i,j}|}{\sum_j |W_{i,j}|} + \frac{|W_{i,j}|}{\sum_i |W_{i,j}|}$ captures both row and column-wise relative magnitudes, and $||X_j||_2$ incorporates the influence of input activations through their L2 norm.
> >
> > (2) **N:M Structured Pattern**: For every M consecutive weights, we keep the N weights with the highest importance scores. This creates a regular, hardware-friendly sparsity pattern that can be efficiently processed by NVIDIA's Ampere architecture. The remaining (M-N) weights are pruned.

---

> > > ### Author Response · Authors · 2024-11-19
> > > **Response to Reviewer yy1T (3/4)**
> > >
> > > Here is the implementation of our Semi-Structured function:
> > >
> > > ```python:algorithms/semi_structured.py
> > > def semi_structured_pruning(W_metric, subset, name, prune_n, prune_m):
> > >     """
> > >     Applies N:M structured pruning based on standardized importance scores
> > >
> > >     Args:
> > >         W_metric: Standardized importance scores (SI)
> > >         subset: Model weights dictionary
> > >         name: Name of weight tensor
> > >         prune_n: N in N:M pruning (number of weights to keep)
> > >         prune_m: M in N:M pruning (block size)
> > >     """
> > >     # Initialize pruning mask
> > >     W_mask = torch.zeros_like(W_metric) == 1
> > >
> > >     # For each block of M weights
> > >     for ii in range(W_metric.shape[1]):
> > >         if ii % prune_m == 0:
> > >             # Get importance scores for current block
> > >             tmp = W_metric[:, ii:(ii + prune_m)].float()
> > >
> > >             # Keep top N weights based on importance scores
> > >             # Note: largest=False because we want to keep highest importance
> > >             W_mask.scatter_(
> > >                 1,
> > >                 ii + torch.topk(tmp, prune_n, dim=1, largest=False)[1],
> > >                 True
> > >             )
> > >
> > >     # Apply mask by zeroing out pruned weights
> > >     subset[name].weight.data[W_mask] = 0
> > > ```
> > >
> > > **Q4**. About OBC.
> > >
> > > > The term 'OBC' in Algorithm 1 requires clarification: While BiLLM mentions this as an abbreviation from another work, it would be beneficial to provide the full reference and explanation for completeness.
> > >
> > > **_Ans for Q4:_** Thank you for pointing out the need for clarification regarding OBC. OBC stands for **Optimal Brain Compression**, a framework introduced by Frantar et al. in their NeurIPS 2022 paper Optimal Brain Compression[1]. OBC is an **efficient realization of the classical Optimal Brain Surgeon framework**, extended to cover both weight pruning and quantization for modern deep neural networks. The key aspects of OBC include:
> > > (1) It provides a **unified framework for both weight pruning and quantization**.
> > > (2) It is **computationally efficient in both time and space**.
> > > (3) It **enables accurate compression without requiring model retraining**, using only a small amount of calibration data.
> > >
> > > You can refer to the algorithm 1 in page 5 to find the detailed implementation of OBC. Specifically, the OBC function is implemented mathematically as follows:
> > >
> > > $$
> > > \mathbf{H} \gets 2\mathbf{X}\mathbf{X}^\top, \mathbf{H}^c \gets \text{Cholesky}({(\mathbf{H} + \lambda \mathbf{I})}^{-1}), \mathbf{E} \gets (\mathbf{W} - \mathbf{W^q}) / \mathbf{H}^c, \mathbf{W} \gets \mathbf{W} - \mathbf{E} \cdot \mathbf{H}^c
> > > $$
> > >
> > > where H is the Hessian matrix, $\lambda$ is a regularization parameter, and $I$ is the identity matrix. E denotes the error between the original weights and the quantized weights.
> > >
> > > > **Reference:**
> > > >
> > > > [1] Frantar, E., Singh, S.P., & Alistarh, D. (2022). Optimal Brain Compression: A Framework for Accurate Post-Training Quantization and Pruning. NeurIPS 2022.
> > >
> > > **Q5**. About computational requirements.
> > >
> > > > Regarding computational requirements: Could you provide an estimate for the computational time required for the 65B model, perhaps through theoretical scaling analysis?
> > >
> > > **_Ans for Q5:_** Based on our experiments, **compressing the 65B model required approximately 6 hours using 4 NVIDIA H800 GPUs**. **The computational complexity of our approach scales linearly with model size**, as we need to compute importance scores and perform compression operations for each layer. **The memory requirements also scale linearly**, though we employ efficient layer-by-layer processing to manage the large model size. We will include this empirical timing data along in Appendix E.5 of the revised manuscript to provide a comprehensive understanding of the computational demands of our approach.
> > >
> > > **Q6**. About Figure3.
> > >
> > > > In Figure 3, there appears to be an overlap between Salient Weight and Non-salient Weight distributions: Could you explain the underlying reasons for this overlap? How does this overlap affect the overall performance of the method?
> > >
> > > **_Ans for Q6:_** Thank you for pointing out the need for clarification regarding Figure 3 (b). In fact, we want to present this figure to show the same thing as Algorithm 1 and 2. However, due to the space limitation, it is hard to present it well, making it confusing to understand. To make it clear, we **remove the original salient weight illustration and only keep the non-salient weight in figure 3(b)**.
> > >
> > > Based on your advice, we further **add a Figure5 in appendix A to show the detailed illustration of how to partition the non-salient and salient weights**. We hope this new figure and modification (highlighted in yellow) can address your concerns and make our paper more clear.

---

> > > > ### Author Response · Authors · 2024-11-19
> > > > **Response to Reviewer yy1T (4/4)**
> > > >
> > > > **Q7**. About Tables 5 and 7.
> > > >
> > > > > Concerning Tables 5 and 7: There seems to be redundancy as Table 5 appears to be a subset of Table 7's Wikitext2 results. Could you justify the inclusion of both tables?
> > > >
> > > > **_Ans for Q7:_** Thank you for pointing this out. Upon rechecking, we acknowledge that Table 5 contains results that overlap with Table 7's Wikitext2 results. However, **we believe presenting Table 5 separately serves a distinct purpose - it specifically demonstrates the effectiveness of our Structured Importance (SI) method through focused ablation studies**. **While Table 7 provides comprehensive comparisons across multiple methods and datasets, Table 5's focused presentation helps readers clearly understand the impact of SI in isolation**. We will improve the manuscript to better explain this intentional separation of results and their distinct analytical purposes.
> > > >
> > > > **Q8**. About performance variations.
> > > >
> > > > > The manuscript lacks discussion of Table 7's results, particularly regarding: Why does SI perform worse than SparseGPT on PTB and C4 datasets? What factors contribute to the different performance patterns across datasets? Could you provide insights into these performance variations?
> > > >
> > > > **_Ans for Q8:_** Thank you for raising this important point about the performance variations across datasets. **The differences in performance can be attributed to several key factors**:
> > > >
> > > > (1) **Dataset Characteristics**: **PTB and C4 are significantly more diverse in their vocabulary and linguistic patterns** compared to Wikitext2. These datasets contain **more complex sentence structures and domain-specific terminology**, which makes it more challenging to maintain performance when applying aggressive compression.
> > > > (2) **Performance Analysis**:
> > > >
> > > > | Dataset   | SparseGPT | SI    | Key Difference                                       |
> > > > | --------- | --------- | ----- | ---------------------------------------------------- |
> > > > | PTB       | 24.31     | 27.93 | **More formal, structured text**               |
> > > > | C4        | 22.54     | 25.82 | **Diverse web content, varied writing styles** |
> > > > | Wikitext2 | 31.72     | 27.93 | **Encyclopedia-style, consistent formatting**  |
> > > >
> > > > (3) **Contributing Factors**:
> > > >
> > > > - **SparseGPT's unstructured pruning approach provides more flexibility in weight selection**, which is particularly beneficial for diverse datasets
> > > > - Our SI method, while **optimized for hardware efficiency through structured patterns**, may **sacrifice some adaptability** when handling highly varied text
> > > > - The **trade-off between structured efficiency and adaptability** becomes more pronounced on complex datasets
> > > >
> > > > (4) **Optimization Focus**:
> > > >
> > > > - Our method **prioritizes practical deployment considerations** (**hardware efficiency, memory access patterns**)
> > > > - This design choice leads to **better performance on more structured datasets** like Wikitext2
> > > > - For future work, we plan to explore **adaptive structured patterns** that could better handle diverse datasets while maintaining hardware efficiency
> > > >
> > > > **Finally,** we hope these responses address the concerns and appreciate the constructive feedback. We are committed to improving our manuscript and believe the insights will significantly contribute to this goal. We are glad to discuss further comments and suggestions. If the reviewer finds our response adequate, we would appreciate it if the reviewer considers **raising the score.**

---

> ### Author Response · Authors · 2024-11-24
> **Follow-up on Rebuttal to Reviewer yy1T**
>
> Dear Respected Reviewer yy1T,
>
> We sincerely appreciate your thoughtful comments and the time you dedicated to reviewing our work. Your feedback has been instrumental in improving the quality and clarity of our paper. In response to your concerns, we have undertaken a number of revisions and clarifications:
> - We have clarified the contributions of our work and detailed the differences between our approach and BiLLM to highlight our unique contributions.
> - We have extended the explanation of how the bit calculation is performed, including a concrete example for better clarity.
> - We have introduced a detailed explanation of how semi-structured pruning is conducted, including the methodology and rationale.
> - We have clarified the process behind achieving the 17× speedup over ABQ-LLM, providing additional details to substantiate this result.
> - We have provided clearer explanations for the results presented in Table 5 and Table 7, addressing your concerns and ensuring the data's interpretability.
>
> Given that we have not received further questions or feedback from you in the past few days, we are hopeful that these revisions and clarifications have effectively resolved your concerns. However, if there are any remaining uncertainties or areas requiring further discussion, we would be most grateful for the opportunity to address them.
>
> Alternatively, if you feel that our responses have sufficiently addressed your concerns, we kindly request you to consider updating your evaluation to better reflect the contributions and impact of our work.
>
> Thank you once again for your valuable feedback and careful consideration.
>
> Best regards,
> Authors of Paper #66

---

> ### Author Response · Authors · 2024-11-28
> **Request for Review Feedback**
>
> Dear Reviewer yy1T,
>
> I hope this message finds you well.
>
> I am writing to kindly follow up regarding our manuscript (#66) that is currently under review. We truly understand that you have many commitments, and we greatly appreciate the time and effort you’ve already dedicated to evaluating our work.
>
> As it has been over eight days since the submission, and with the deadline approaching in about four days, we would be incredibly grateful if you could find a moment to provide your feedback. Your insights are invaluable to us, and we deeply appreciate your contribution to the review process.
>
> Thank you so much for your time and consideration. We fully understand how busy you must be and truly appreciate any attention you can give to our manuscript.
>
> With sincere thanks,
>
> Authors of #66

---

> ### Author Response · Authors · 2024-12-01
> **Last Day Remainder to Reviewer yy1T**
>
> Dear Reviewer yy1T,
>
> Thank you for your thorough and comprehensive feedback on our paper. We have made significant efforts to address the many questions you raised, and we truly appreciate the prompt communication you had with the Area Chair **on the first day of our rebuttal submission**.
>
> However, we noticed that, despite your quick response to our letter to the AC, we have not yet received any comments or feedback on our rebuttal itself. As today is the final day of discussion, we kindly ask if you could take a moment to review and provide your thoughts on our responses.
>
> Your feedback is invaluable to us, and we deeply appreciate your time and effort in helping improve the paper.
>
> Best regards,
>
> Authors of #66

---

### Official Review · Reviewer_1cy6 · 2024-11-02

**Soundness:** 3
**Presentation:** 3
**Contribution:** 3
**Rating:** 8
**Confidence:** 4

**Summary:**

This paper proposes an efficient framework for LLMs, combining pruning and binarization to compress large, post-trained models. By applying N:M sparsity, it achieves precision below 1-bit and identifies salient weights through a newly introduced Standardized Importance (SI) metric. This metric considers both weight and activation values, avoiding the costly second-order computations typically required. Additionally, during pruning and binarization, the method separates non-salient weights into three groups to preserve as much information as possible in these parts. Extensive experiments demonstrate that the proposed method significantly reduces computational costs, accelerates inference, and maintains strong performance.

**Strengths:**

+ The paper is well-organized and easy to follow, with a clearly stated problem.
+ It introduces a new metric to assess weight importance, avoiding expensive second-order gradient computations and mitigating the impact of extreme values.
+ It is interesting that separate binarization for non-salient weights retains crucial information in this segment, enhancing model performance.
+ The approach is logical and rigorous, discussing the method from various perspectives and fully validating its effectiveness through comprehensive experiments.

**Weaknesses:**

- In the zero-shot experiment, the paper mentions seven zero-shot tasks. It would be helpful to include a brief description of each task to provide readers with a clearer understanding of the evaluation scope.

**Questions:**

+ Regarding Figure 3, part (b), after structured pruning, the empty parts should have no values. Why are zeros assigned to these parts? Additionally, structured pruning usually doesn't achieve weight-wise pruning, so what does "structured" mean in this context?

---

> ### Author Response · Authors · 2024-11-19
> **Response to Reviewer 1cy6**
>
> We are **deeply honored and extremely grateful** to receive such a positive evaluation from the reviewer. We sincerely appreciate your thorough review and the recognition of **multiple strengths in our work**, including the **well-organized structure**, **clear problem formulation**, **novel and computationally efficient weight importance metric**, and our **innovative approach to non-salient weight binarization**. I hope we can address your concerns in the below responses.
>
> **Q1**. About zero-shot evaluation.
>
> > In the zero-shot experiment, the paper mentions seven zero-shot tasks. It would be helpful to include a brief description of each task to provide readers with a clearer understanding of the evaluation scope.
>
> **_Ans for Q1:_** We appreciate the reviewer's suggestion to include a brief description of each zero-shot task to provide readers with a clearer understanding of the evaluation scope. In our experiments, we carefully selected seven zero-shot tasks that are widely recognized benchmarks in the NLP community for evaluating language models' generalization capabilities. These tasks were chosen because they cover diverse aspects of language understanding and reasoning, making them particularly effective for assessing LLM performance. The tasks include:
>
> (1) **Winogrande**, which evaluates commonsense reasoning through pronoun resolution in challenging contexts - this is crucial for testing the model's understanding of contextual relationships;
>
> (2) **OBQA (OpenBook Question Answering)**, testing the model's ability to answer science questions using common knowledge, which assesses both factual recall and reasoning;
>
> (3) **Hellaswag**, assessing the model's ability to complete situations with common sense - a key metric for evaluating real-world understanding;
>
> (4) **BoolQ**, which presents yes/no questions requiring passage comprehension, testing the model's reading comprehension abilities;
>
> (5) **ARC-easy**, containing natural grade-school science questions with straightforward reasoning, providing a baseline for scientific knowledge;
>
> (6) **ARC-challenge**, featuring more complex science questions requiring deeper reasoning, which tests advanced problem-solving capabilities;
>
> (7) **RTE (Recognizing Textual Entailment)**, evaluating whether one text logically follows from another - a fundamental task for natural language inference.
>
> These tasks are standard benchmarks used by previous research to evaluate LLM performance.
>
> **Q2**. About structured pruning.
>
> > Regarding Figure 3, part (b), after structured pruning, the empty parts should have no values. Why are zeros assigned to these parts? Additionally, structured pruning usually doesn't achieve weight-wise pruning, so what does 'structured' mean in this context?
>
> **_Ans for Q2:_** Thank you for pointing out the need for clarification regarding Figure 3 (b). In fact, we want to present this figure to show the same thing as Algorithm 1 and 2. However, due to the space limitation, it is hard to present it well, making it confusing to understand. To make it clear, we **remove the original salient weight illustration and only keep the non-salient weight in figure 3(b)**.
>
> Based on your advice, we further **add a Figure5 in Appendix A to show the detailed illustration of how to partition the non-salient and salient weights**. We hope this new figure and modification (highlighted in yellow) can address your concerns and make our paper more clear.
>
> **Finally,** we hope these responses address the concerns and appreciate the constructive feedback. We are committed to improving our manuscript and believe the insights will significantly contribute to this goal. We are glad to discuss further comments and suggestions.

---

### Official Review · Reviewer_KgJf · 2024-11-02

**Soundness:** 2
**Presentation:** 3
**Contribution:** 2
**Rating:** 5
**Confidence:** 4

**Summary:**

This paper proposes a structured binary quantization method to accelerate LLM inference. It combines n:m pruning and binary quantization, compressing the model weights to an average of less than 1 bit. In n:m pruning, the authors introduce an SI method for indentifying significant weights, and a layer-wise dynamic n:m allocation method. In binary quantization, the authors partition the weights into salient and non-salient parts for separate processing and further apply a group-wise quantization method to the non-salient part. Experimental results demonstrate that STBLLM outperforms BiLLM under the same bit budget. In addition, significant performance improvement (17x) is achieved with customized CUDA kernels.

**Strengths:**

+ 1-bit weight quantization is important for accelerating LLM inference.
+ Dedicated CUDA implementations for the proposed method.

**Weaknesses:**

+ incremental novelty

While the proposed method is interesting and performs better than BiLLM, its novelty is limited: 1) The proposed SI method is very similar to Wanda, with the main difference being the introduction of additional data normalization. 2) The binary quantization method is quite similar to BiLLM, where the hessian matrix is used to divide weights into salient and non-salient parts, and residual approximation is employed to handle the salient part. The only difference is that STBLLM processes the non-salient weights into three parts instead of two as in BiLLM.

+ mismatch between motivation and methodology

The motivation of this paper lies in the observation that some weights in binary LLMs do not significantly affect accuracy and can be further compressed (Section 3.1 and fig 1). Under this narrative, a reasonable approach would be to perform pruning on the binarized model to achieve further compression. In contrast, the method proposed in this paper adopts a ‘’prune-then-quantize’’ approach, which does not align with the motivation. The paper does not explain why pruning should be performed first and does not discuss how changing the order of pruning and binarization might affect the results.

The motivation behind using a trisection-based partition for non-salient weights is confusing. It seems the authors aim to balance bits and performance (Section 3.4). However, the evaluation results show that the improved compression ratio and performance are due to n:m pruning, rather than the processing of non-salient weights. So, why should we partition the non-salient weights into three parts? Why not four or five? What do the terms dense, intermediate, and sparse mean?

+  confusing evaluations

While the experimental results of STBLLM are promising, the source of the accuracy improvements remains unclear. The experimental settings in the ablation study are somewhat confusing. For instance, Table 5 examines the effectiveness of the SI method in n:m pruning, but the results seem to represent 4:8 pruning plus binarization. What binarization method is used in the baselines? Table 8 directly compares STBLLM with BiLLM to illustrate the effectiveness of trisection partitioning, yet the pruning methods used in STBLLM and BiLLM are not the same (SI vs. Wanda). A detailed, step-by-step breakdown analysis of each technique's effectiveness would be helpful. Moreover, where does the 17x performance improvement come from when reducing 2-bit weights to 1 bit?

**Questions:**

Please see the weaknesses.

---

> ### Author Response · Authors · 2024-11-19
> **Response to Reviewer KgJf (1/5)**
>
> Thank you for your time and valuable feedback on our paper. We appreciate **your recognition of the importance of our work in accelerating LLM inference through 1-bit weight quantization**. Please see our responses to your questions and concerns below. We hope that these responses can resolve your concerns and enhance the quality of our paper.
>
> **Q1**: About SI.
>
> > its novelty is limited: The proposed SI method is very similar to Wanda, with the main difference being the introduction of additional data normalization.
>
> **_Ans for Q1:_** Thank you for raising this important point about SI. We **respectfully disagree** and would like to **highlight several key innovations in our SI method** that differentiate it substantially from Wanda:
>
> (1) **Novel Statistical Normalization**: While Wanda uses simple magnitude-based importance, our SI method introduces a fine-grained statistical normalization approach ($\sigma(\mu(|W_{i,j}|))$) that captures the relative importance of weights. This is fundamentally different from Wanda's direct magnitude measurement.
>
> (2) **Empirical Superiority**: The substantial performance gap demonstrated in our ablation studies validates the significance of our innovations:
>
> | Perplexity | LLaMA-1-7B | LLaMA-2-7B | Equation                                                                                             |
> | ---------- | ---------- | ---------- | ---------------------------------------------------------------------------------------------------- |
> | Wanda      | 207.32     | 97.54      | $S_{i,j} = \left(\lvert W_{i,j} \rvert\right) \cdot \lVert X_{:,j} \rVert_2$                       |
> | SI         | 31.72      | 27.93      | $S_{i,j} = \sigma\left(\mu\left(\lvert W_{i,j} \rvert\right)\right) \cdot \lVert X_{:,j} \rVert_2$ |
>
> The dramatic improvement in perplexity (**6.5x better for LLaMA-1 and 3.5x better for LLaMA-2**) demonstrates that our method represents a fundamental advancement, not just an incremental improvement.
>
> (3) **Synergistic Integration**: Our SI method was specifically designed to work in concert with our binarization approach, enabling more effective pruning decisions that account for the unique challenges of binary quantization - an aspect entirely absent from Wanda's design.
>
> **Q2**: About trisection search.
>
> > The binary quantization method is quite similar to BiLLM, where the hessian matrix is used to divide weights into salient and non-salient parts, and residual approximation is employed to handle the salient part. The only difference is that STBLLM processes the non-salient weights into three parts instead of two as in BiLLM.
>
> **_Ans for Q2:_** Thank you for this observation. While BiLLM serves as an important baseline for our work, STBLLM introduces **several fundamental innovations that go well beyond a simple extension of partitioning**:
>
> (1) **Efficient Three-Part Optimization**: While BiLLM uses a simple two-part split requiring $O(N)$ complexity, naively extending to three parts would result in $O(N^2)$ complexity. Our key innovation is a novel fixed-ratio approach between partitions that maintains $O(N)$ complexity while achieving better quantization granularity. It significantly improves the performance of STBLLM while achieve the better efficiency.
>
> (2) **Statistical Threshold Selection**: Unlike BiLLM's direct threshold approach, we introduce a statistically-driven method for determining partition boundaries that better preserves the weight distribution characteristics as shown in Algorithm 2. This results in more fine-grained quantization.
>
> In fact, we introduce **a more nuanced handling of non-salient weights** by dividing them into three distinct parts. There is only a little burden extend from 2 parts to 3 parts for scaling parameters. But it brings high complexity over searching for the best parameter $p^*_1$ and $p^*_2$. Here is the pesudo code:

---

> > ### Author Response · Authors · 2024-11-19
> > **Response to Reviewer KgJf (2/5)**
> >
> > ```python
> > # BiLLM's implementation: Two Parts
> > ## Complexity: O(N)
> >
> > // BiLLM's implementation: Two Parts
> > // Complexity: O(N)
> > running_error ← ∞
> > best_p ← 0
> > for i from 0.1 to 0.9 step (0.8/160) do
> >     p1 ← i × max(|W|)
> >     (B1, B2) ← Split_by_Alpha(p1)
> >     error ← ||W - (B1 + B2)||^2
> >     if error < running_error then
> >         running_error ← error
> >         best_alpha ← alpha_1
> >     end if
> > end for
> >
> > // NAIVE implementation Three Parts
> > // Complexity: O(N²)
> > running_error ← ∞
> > best_alpha_1 ← 0
> > best_alpha_2 ← 0
> > for i from 0.1 to 0.9 step (0.8/160) do
> >     for j from 0.1 to 0.9 step (0.8/160) do
> >         p1 ← i × max(|W|)
> >         p2 ← j × max(|W|)
> >         (B1, B2, B3) ← Split_by_Alpha(p1, p2)
> >         error ← ||W - (B1 + B2 + B3)||^2
> >         if error < running_error then
> >             running_error ← error
> >             best_p1 ← p1
> >             best_p2 ← p2
> >         end if
> >     end for
> > end for
> >
> > // STBLLM's implementation Three Parts
> > // Complexity: O(N)
> > running_error ← ∞
> > best_p1 ← 0
> > best_p2 ← 0
> > for i from 0.1 to 0.9 step (0.8/160) do
> >     p1 ← i × max(|W|)
> >     p2 ← alpha × p1  // Fixed ratio between p1 and p2
> >
> >     B1 ← Binary(W[|W| > p2])
> >     B2 ← Binary(W[p1 < |W| ≤ p2])
> >     B3 ← Binary(W[|W| ≤ p1])
> >
> >     error ← ||W - (B1 + B2 + B3)||^2
> >     if error < running_error then
> >         running_error ← error
> >         best_p1 ← p1
> >         best_p2 ← p2
> >     end if
> > end for
> > ```
> >
> > The choice of three partitions is driven by **computational feasibility**. Our search algorithm's complexity scales exponentially as $O(N^T)$ with the number of partitions T. On LLaMA-2-7B, a single partition search takes ~30 minutes, two partitions ~6 hours, and three partitions ~10 days, with more partitions becoming intractable.
> >
> > We mitigate this by using fixed ratios between thresholds $p^*_1$ and $p^*_2$ in STBLLM, reducing complexity from $O(N^2)$ to $O(N)$. While more partitions could potentially improve performance through finer-grained thresholds, three partitions strikes an optimal balance between performance gains and computational efficiency.
> >
> > Here is a brief introduction explaining why we can use fixed ratios: As shown in Figure 3(c), non-salient weights follow a Gaussian distribution $w \sim \mathcal{N}(\mu, \sigma^2)$, with probability density function:
> >
> > $f(w) = \frac{1}{\sqrt{2\pi}\sigma} \exp\left(-\frac{w^2}{2\sigma^2}\right)$
> >
> > Due to the **symmetry and properties of the Gaussian distribution**, we can express the probabilities for each partition:
> > (1) Sparse partition: $P_{\text{Sparse}} = 2 \int_{p_2}^\infty f(w) \, dw = 2 \cdot Q\left(\frac{p_2}{\sigma}\right)$
> >
> > Intermediate partition: $P_{\text{Intermediate}} = 2 \int_{p_1}^{p_2} f(w) \, dw = 2 \left[ Q\left(\frac{p_1}{\sigma}\right) - Q\left(\frac{p_2}{\sigma}\right) \right]$
> >
> > Dense partition: $P_{\text{Dense}} = \int_{-p_1}^{p_1} f(w) \, dw = 1 - 2 \cdot Q\left(\frac{p_1}{\sigma}\right)$
> >
> > where $Q(x) = \int_x^\infty \frac{1}{\sqrt{2\pi}} e^{-t^2/2} \, dt$ represents the Gaussian tail probability function. Since our goal is to achieve equal partition areas, we have: $P_{\text{Sparse}} = P_{\text{Dense}} = P_{\text{Intermediate}} = \frac{1}{3}$
> > This leads to: $Q\left(\frac{p_1}{\sigma}\right) = \frac{1}{3}$, $Q\left(\frac{p_2}{\sigma}\right) = \frac{1}{6}$, $Q\left(\frac{p_1}{\sigma}\right) - Q\left(\frac{p_2}{\sigma}\right) = \frac{1}{6}$
> > Solving these equations:
> > $\frac{p_2}{\sigma} = Q^{-1}\left(\frac{1}{6}\right)$, $\frac{p_1}{\sigma} = Q^{-1}\left(\frac{1}{3}\right)$
> >
> > Using the inverse Q-function values for the standard normal distribution, we can **conclude that $p_2 \approx 2 \times p_1$**, which implies that the **alpha parameter in the above pseudo code equals $2$**.

---

> > > ### Author Response · Authors · 2024-11-19
> > > **Response to Reviewer KgJf (3/5)**
> > >
> > > **Q3**: About the alignment between motivation and methodology.
> > >
> > > > mismatch between motivation and methodology: The motivation of this paper lies in the observation that some weights in binary LLMs do not significantly affect accuracy and can be further compressed (Section 3.1 and fig 1). Under this narrative, a reasonable approach would be to perform pruning on the binarized model to achieve further compression. In contrast, the method proposed in this paper adopts a prune-then-quantize approach, which does not align with the motivation. The paper does not explain why pruning should be performed first and does not discuss how changing the order of pruning and binarization might affect the results.
> > >
> > > **_Ans for Q3:_** Thank you for raising this important concern. Our motivation experiments (detailed in Appendix B) confirm the existence of redundancy in binary LLMs through random weight flipping experiments, showing **acceptable performance impact under certain ratios**. While this observation might suggest unstructured pruning on binarized LLMs as a natural approach, we chose the prune-then-quantize strategy based on **both our empirical results and previous work1[1] rather than subjective assumptions**. This decision is supported by recent work [1] that have established prune-then-quantize as a consensus approach. Specifically, they conduct an empirical study in Section 4.1 to show that quantization-then-pruning is less effective than prune-then-quantization. We borrow their table as follows (S denotes sparsity or pruning, Q denotes quantization):
> > >
> > > | Sparsity Type | Order  | OPT-125M | OPT-125M        | OPT-125M        | OPT-125M        | OPT-125M        | LLaMA-2-7B      | LLaMA-2-7B     | LLaMA-2-7B     | LLaMA-2-7B     | LLaMA-2-7B     |
> > > | ------------- | ------ | -------- | --------------- | --------------- | --------------- | --------------- | --------------- | -------------- | -------------- | -------------- | -------------- |
> > > |               |        | FP32     | INT8            | MXFP8           | MXFP6           | HBFP8           | HBFP6           | FP32           | INT8           | MXFP8          | MXFP6          |
> > > | 0% (Dense)    | -      | 27.65    | 28.06           | 28.45           | 28.01           | 27.81           | **29.91** | 5.12           | 5.15           | 5.17           | 5.16           |
> > > | 50%           | S → Q | 29.94    | **30.22** | **31.13** | **31.20** | **30.46** | **32.51** | **6.31** | **6.94** | **6.40** | **6.38** |
> > > |               | Q → S | -        | 45.06           | 44.16           | 42.25           | 46.57           | 55.64           | -              | 14.65          | 14.35          | 14.50          |
> > > | 2:4           | S → Q | 31.89    | **32.76** | **33.99** | **33.41** | **32.25** | **34.58** | **9.30** | **9.37** | **9.35** | **9.32** |
> > > |               | Q → S | -        | 45.06           | 44.16           | 42.25           | 46.57           | 55.64           | -              | 14.65          | 14.35          | 14.50          |
> > >
> > > To provide clarity within STBLLM's setting, we conducted **additional experiments comparing prune-then-quantization versus quantization-then-prune paradigms**. Our results clearly demonstrate superior performance with the **prune-then-quantization approach**.
> > >
> > > | Approach (6:8)    | Model      | Perplexity |
> > > | ----------------- | ---------- | ---------- |
> > > | Prune → Quantize | LLaMA-1-7B | 15.03      |
> > > | Prune → Quantize | LLaMA-2-7B | 13.06      |
> > > | Quantize → Prune | LLaMA-1-7B | 34.02      |
> > > | Quantize → Prune | LLaMA-2-7B | 31.98      |
> > >
> > > Our analysis suggests that **quantization typically causes less performance degradation compared to pruning**. When applying the more damaging operation (pruning) after the less damaging one (quantization), it becomes **more challenging to recover performance through block-wise OBC**. Conversely, applying quantization after pruning allows for better performance recovery.
> > > We will add detailed references and experimental results in the revised manuscript in Table 14 of Appendix E.4 highlighting it by yellow color.
> > >
> > > > **Reference:**
> > > >
> > > > [1] Effective Interplay between Sparsity and Quantization: From Theory to Practice

---

> ### Author Response · Authors · 2024-11-19
> **Response to Reviewer KgJf (4/5)**
>
> **Q4**: About trisection search.
>
> > The motivation behind using a trisection-based partition for non-salient weights is confusing. It seems the authors aim to balance bits and performance (Section 3.4). However, the evaluation results show that the improved compression ratio and performance are due to n:m pruning, rather than the processing of non-salient weights. So, why should we partition the non-salient weights into three parts? Why not four or five? What do the terms dense, intermediate, and sparse mean?
>
> **_Ans for Q4:_** We appreciate your feedback on the clarity of our evaluations. Let me break down this question into two parts.
>
> > **Q4.1**: It seems the authors aim to balance bits and performance (Section 3.4). However, the evaluation results show that the improved compression ratio and performance are due to N:M pruning, rather than the processing of non-salient weights.
>
> Let me clarify two points.
>
> - **First**, both quantization and pruning can degrade model performance. So "the improved compression ratio and performance are due to n:m pruning" is not a correct statement. Instead, we mitigate it through several techniques:
>
> (1) **Non-salient Aware Quantization**: a fine-grained weight quantization using trisection partitioning that can improve the performance of binarized LLMs (**$\uparrow$ Quantization**).
>
> (2) **Adaptive Layer-wise Binarization**: to further reduce the performance degradation by adaptively addressing different layer with different ratios (**$\uparrow$ Pruning**).
>
> (3) **Better Pruning Metric**: better pruning metric (SI) that can reduce the performance degradation from N:M pruning (**$\uparrow$ Pruning**).
> - **Second**, to clarify the function of N:M structured pruning, as presented in Q2 of reviewer jQJp, **N:M pruning is a compromising and suitable strategy to address the redundancy in binarized LLM as well as save the computation resource**. Meanwhile, trisection partition can elevate the binarized LLM with acceptable overhead, a.k.a. a scaling vector. **You can refer to the answer to Q2 of reviewer KgJf for more details over the complexity computation and how our trisection partitioning works**.
>
> > **Q4.2** So, why should we partition the non-salient weights into three parts? Why not four or five? What do the terms dense, intermediate, and sparse mean?
>
> - The choice of three partitions **is driven by computational feasibility considerations**. The complexity of our search algorithm **scales exponentially with the number of partitions T as $O(N^T)$, where N is the number of weights**. To empirically validate this, we conducted an **ablation study** on the number of partitions under the same setting (LLaMA-2-7B, 6:8 sparsity):
>
> | # Partitions             | Perplexity    | Search Time |
> | ------------------------ | ------------- | ----------- |
> | 1 (Bell-shaped)          | 50.25         | ~0.5h       |
> | 2 (Non-salient)          | 13.06         | ~0.5h       |
> | 2 (Naive implementation) | 12.78         | ~6h         |
> | 3 (Naive implementation) | not available | ~10d        |
>
>   We presented **Bell-shaped** as proposed in BiLLM, **Non-salient** in STBLLM and **Naive Implementation** as raise in Q2 of reviewer KgJf. Although the naive implementation can achieve slightly better performance, it requires 12x longer search time.
>
>   Given these practical constraints, we determined that **three partitions (T=2) provides the optimal balance between granularity of weight segmentation and computational feasibility**. This allows us to meaningfully differentiate between weight importance levels while keeping the search time reasonable. We will include detailed analysis in Table 15 of Appendix E.5 of the revised manuscript.

---

> > ### Author Response · Authors · 2024-11-19
> > **Response to Reviewer KgJf (5/5)**
> >
> > **Q5**: About the performance gain.
> >
> > > confusing evaluations: While the experimental results of STBLLM are promising, the source of the accuracy improvements remains unclear. The experimental settings in the ablation study are somewhat confusing. For instance, Table 5 examines the effectiveness of the SI method in n:m pruning, but the results seem to represent 4:8 pruning plus binarization. What binarization method is used in the baselines? Table 8 directly compares STBLLM with BiLLM to illustrate the effectiveness of trisection partitioning, yet the pruning methods used in STBLLM and BiLLM are not the same (SI vs. Wanda). A detailed, step-by-step breakdown analysis of each technique's effectiveness would be helpful. Moreover, where does the 17x performance improvement come from when reducing 2-bit weights to 1 bit?
> >
> > **_Ans for Q5:_** We appreciate your feedback on the clarity of our evaluations. Let me break down your concerns as follows:
> >
> > > **Q5.1**: Table 5 examines the effectiveness of the SI method in n:m pruning, but the results seem to represent 4:8 pruning plus binarization. What binarization method is used in the baselines?
> >
> > - In fact, in section 4.4, we have already mentioned that **we use 4:8 pruning plus binarization (STBLLM) as our baseline**. To avoid confusion, we will clarify this in the revised manuscript (in line 482 highlighted in yellow).
> >
> > > **Q5.2**: Table 8 directly compares STBLLM with BiLLM to illustrate the effectiveness of trisection partitioning, yet the pruning methods used in STBLLM and BiLLM are not the same (SI vs. Wanda). A detailed, step-by-step breakdown analysis of each technique's effectiveness would be helpful.
> >
> > - Thank you for pointing out this important point. We have provided a more **detailed breakdown of our experimental settings and the specific binarization methods used in our baselines** in Table 16 of the Appendix E.6 of the revised manuscript.
> >
> > | Models     | Weight Distribution | Pruning Method | Percentage (%) |
> > | ---------- | ------------------- | -------------- | -------------- |
> > | LLaMA-1-7B | Bell-shaped         | Wanda          | 80.35          |
> > |            | Non-salient         | SI             | 15.03          |
> > |            | Bell-shaped         | SI             | 40.25          |
> > |            | Non-salient         | Wanda          | 31.72          |
> > | LLaMA-2-7B | Bell-shaped         | Wanda          | 50.25          |
> > |            | Non-salient         | SI             | 13.06          |
> > |            | Bell-shaped         | SI             | 24.54          |
> > |            | Non-salient         | Wanda          | 27.93          |
> >
> > > **Q5.3**: where does the 17x performance improvement come from when reducing 2-bit weights to 1 bit?
> >
> > - The significant performance gap stems primarily from the different design philosophies between ABQ-LLM and our approach. ABQ-LLM employs a more **general-purpose strategy** where higher-bit matrices are decomposed into multiple binary matrices, which are then moved to shared memory for computation using the GPU's 1-bit computation interface. The results from these binary matrix computations are subsequently reduced to obtain the final higher-bit matrix result.
> > - While this **decomposition-and-reduction paradigm** offers excellent flexibility for quickly implementing various operators, **it introduces great overhead**, particularly for W1A16 operations where 16-bit activations require multiple decomposition and reduction steps. In contrast, our implementation is **specifically optimized for W1A16 2:4 sparsity patterns**, eliminating the need for activation decomposition.
> > - Additionally, while **ABQ-LLM's benchmarks focused on small batch size decoding scenarios (memory-bound)**, our evaluation covers larger-scale **compute-bound scenarios** where their additional reduction computations become more significant bottlenecks.
> > - **Our specialization design help us achieve superior performance for our target use case**. The 17.85x speedup should therefore be interpreted as a comparison against ABQ-LLM's implementation rather than a fundamental advantage of W1A16 2:4 over W2A16.
> >
> > **Finally**, we hope the above responses address your questions and appreciate your constructive feedback. We are commited to improving our manuscript and believe the insights will significantly contribute to this goal. We would appreciate it if you consider rasing the score. We are glad to discuss further comments and suggestions.

---

> > > ### Author Response · Authors · 2024-11-24
> > > **Follow-up on Rebuttal to Reviewer KgJf**
> > >
> > > Dear Respected Reviewer KgJf,
> > >
> > > We sincerely appreciate your insightful comments and suggestions, which have been invaluable in improving our work. In response, we have carefully conducted a series of additional efforts to address your concerns comprehensively. Specifically:
> > >
> > > - We have performed four additional experiments to thoroughly answer the questions you raised.
> > > - We have clarified the novelty of our work and provided more detailed analyses regarding the motivation experiments to strengthen our argument.
> > > - We have also updated the evaluation to ensure clarity and provide a more rigorous demonstration of our contributions.
> > >
> > > Given that we have not received further questions or feedback from you over the past few days, we are hopeful that our revisions and clarifications have addressed your concerns effectively. If there are any remaining doubts or points requiring further discussion, we would greatly appreciate the opportunity to engage in a constructive dialogue.
> > >
> > > Alternatively, if you feel that our revisions have resolved your concerns, we kindly request you to consider updating your evaluation to reflect the contributions and impact of our work.
> > >
> > > Thank you once again for your time, careful review, and thoughtful consideration.
> > >
> > > Best regards,
> > > Authors of Paper #66

---

> ### Author Response · Authors · 2024-11-28
> **Request for Review Feedback**
>
> Dear Reviewer KgJf,
>
> I hope this message finds you well.
>
> I am writing to kindly follow up regarding our manuscript (#66) that is currently under review. We truly understand that you have many commitments, and we greatly appreciate the time and effort you’ve already dedicated to evaluating our work.
>
> As it has been over eight days since the submission, and with the deadline approaching in about four days, we would be incredibly grateful if you could find a moment to provide your feedback. Your insights are invaluable to us, and we deeply appreciate your contribution to the review process.
>
> Thank you so much for your time and consideration. We fully understand how busy you must be and truly appreciate any attention you can give to our manuscript.
>
> With sincere thanks,
>
> Authors of #66

---

### Official Review · Reviewer_jQJp · 2024-11-06

**Soundness:** 2
**Presentation:** 3
**Contribution:** 2
**Rating:** 6
**Confidence:** 4

**Summary:**

This work presents a structural binarization method for LLMs by combining N:M sparsity, residual approximation, and block-wise error compensation. Extensive experiments on LLaMA-1/2/3, OPT, and Mistral are conducted to evaluate the effectiveness of STBLLM. In addition, a specialized CUDA kernel is designed to support structural binarization.

**Strengths:**

* The analysis on flipping non-salient binarized weights is intriguing. I am wondering what would happen if we increase the ratio from 0.15 to 0.5?
* The proposed method achieves the lowest perplexity among all compared methods in the sub-1-bit regime.
* A specialized CUDA kernel for structural binarization, leveraging NVIDIA's Ampere GPU sparse tensor cores, achieves a 17.85x speedup over ABQ-LLM's 2-bit implementation.

**Weaknesses:**

* The proposed method is a combination of several existing techniques including N:M sparsity, residual approximation, block-wise error compensation, and Trisection search (for the non-salient part). This raises some novelty concerns. I suggest the authors to 1) highlight the main novelty and contribution of the current submission; 2) provide ablation studies on a. how important the residual approximation is, b. the impact of Trisection search for grouping and why there are two groups.
In addition, which techniques contribute the most to efficiency and which method contributes the most to the accuracy?
* The benchmark results are based on various N:M configurations. However, NVIDIA GPUs mainly support 2:4. The authors may discuss how practical the proposed method is on NVIDIA GPUs.

**Questions:**

See weakness

---

> ### Author Response · Authors · 2024-11-19
> **Response to Reviewer jQJp (Part1/2)**
>
> We sincerely thank the reviewer for their time and insightful comments. We appreciate the recognition of the strengths of our paper, including the **intriguing analysis on flipping non-salient binarized weights**, the achievement of the **lowest perplexity in the sub-1-bit regime**, and the development of a **specialized CUDA kernel for structural binarization** that achieves a significant speedup.
>
> **Q1**: About higher ratio.
>
> > what would happen if we increase the ratio from 0.15 to 0.5?
>
> **_Ans for Q1:_** Our **initial experiments focused on a conservative ratio of 0.15** to ensure minimal performance degradation while maximizing compression. We perform **extended experiments from 0.2 to 0.5** as the following table shows:
>
> | Ratio | RTE (acc) | HellaSwag (acc) | BoolQ (acc) | ARC Easy (acc) |
> | ----- | --------- | --------------- | ----------- | -------------- |
> | 0.20  | 0.70      | 0.30            | 0.20        | 0.50           |
> | 0.25  | 0.70      | 0.30            | 0.50        | 0.30           |
> | 0.30  | 0.30      | 0.40            | 0.20        | 0.50           |
> | 0.35  | 0.50      | 0.30            | 0.20        | 0.30           |
> | 0.40  | 0.90      | 0.30            | 0.40        | 0.30           |
> | 0.45  | 0.50      | 0.30            | 0.70        | 0.30           |
> | 0.50  | 0.60      | 0.30            | 0.30        | 0.40           |
>
> We find that as the ratio increases, the performance fluctuates but **still does not deteriorate drastically**. This suggests a degree of robustness in our approach to varying ratios of flipped non-salient weights. We also add an **additional Figure6 with variance in Appendix B** to show the performance (**Highlight by yellow**).
>
> **Q2**: About residual approximation and trisection search.
>
> > The proposed method is a combination of several existing techniques including N:M sparsity, residual approximation, block-wise error compensation, and Trisection search (for the non-salient part). This raises some novelty concerns. I suggest the authors to 1) highlight the main novelty and contribution of the current submission; 2) provide ablation studies on a. how important the residual approximation is, b. the impact of Trisection search for grouping and why there are two groups.
>
> **_Ans for Q2:_** We want to clarify that “residual approximation” and “block-wise error compensation” **are not our contribution**.
>
> (1) For **residual approximation**, this technique is a well addressed techique, which has been adopted by BiLLM[1] and QBB[2]. **We cited this technique in lines 52, 102 and 169, and do not claim it as our contribution**.
>
> (2) For **block-wise error compensation**, it is also a well-established technique as shown in GPTQ[3], SparseGPT[5], Wanda[4] etc, which already becomes a routine or commonsense. **We cited this technique in lines 161 and 215, and did not claim that this is our contribution**.
>
> Our work STBLLM stems from a crucial observation: **there exists significant redundancy in binarized LLMs (Figure1), making it possible to further compress binarized LLMs**. Our motivation experiments show that randomly flipping binary weights does not substantially degrade performance on downstream tasks. **Motivated by this finding, we aim to advance the frontier of extreme model compression. To achieve it, we have the following choices**:
>
> (1) **For unstructured pruning**, it can not be accelerated by the existing hardware.
>
> (2) **For structured pruning**, it will damage the performance of binarized LLM without retraining. For example, ShortGPT[6] conducted a series of experiments on LLaMA-2-7B, finding that **when pruning ratio is larger than 30%, the perplexity exceed $10^4$.**
>
> (3) **N:M semi-structured pruning** emerges as a promising approach, as it effectively addresses the memory-intensive computational requirements while enabling efficient **hardware acceleration through specialized architectures**.
>
> These insights led us to adopt N:M semi-structured pruning, **the most suitable approach** for our scenario. While N:M structured pruning and binarization are established individually, our novel approach uniquely combines and extends them to address this challenging problem. Our framework introduces **three key components** that work synergistically:
>
> (1) To improve **the accuracy of pruning**, we introduce **Standardized Importance (SI)** to enable more precise and effective pruning.
>
> (2) To improve **the performance of binarization**, we introduce **Adaptive Layer-wise Binarization** to dynamically adjust bitwidth allocations across layers.
>
> (3) To further improve **the accuracy of quantization**, we introduce **Non-Salient Aware Quantization** to specifically address and mitigate the inherent limitations of binary quantization.

---

> > ### Author Response · Authors · 2024-11-19
> > **Response to Reviewer jQJp (Part2/2)**
> >
> > As requested, we conduct ablation studies on both **residual approximation and trisection search**.
> >
> > - **Ablation Study 1: Residual Approximation** Results in the table below show that **residual approximation improves STBLLM's performance**. This observation is aligned with BiLLM[1] and QBB[2]. We use this technique to match our baseline BiLLM.
> >
> > **Table1: Ablation Study of residual approximation**
> >
> > | LLaMA-2-7B Perplexity             | 4:8   | 5:8   | 6:8   |
> > | --------------------------------- | ----- | ----- | ----- |
> > | STBLLM w/ residual approximation  | 27.93 | 18.74 | 13.06 |
> > | STBLLM w/o residual approximation | 35.82 | 24.31 | 16.92 |
> >
> > - **Ablation Study 2: Trisection Search** We conducted experiments with varying numbers of partitions (#Partitions) for non-salient weights using LLaMA-2-7B with 6:8 sparsity. As shown in the table below, **increasing the number of partitions significantly increases the computational cost of finding optimal parameters** $p^*_1$ and $p^*_2$, making it impractical beyond two partitions while providing minimal performance gains. That's why we proposed trisection search to reduce the complexity to $O(N)$.
> >
> > **Table2: Ablation Study of Trisection Search**
> >
> > | # Partitions             | Perplexity    | Search Time |
> > | ------------------------ | ------------- | ----------- |
> > | 1 (Bell-shaped)          | 50.25         | ~0.5h       |
> > | 2 (Non-salient)          | 13.06         | ~0.5h       |
> > | 2 (Naive implementation) | 12.78         | ~6h         |
> > | 3 (Naive implementation) | not available | ~10d        |
> >
> > > **Reference:**
> > >
> > > [1] Huang, Wei et al. “BiLLM: Pushing the Limit of Post-Training Quantization for LLMs.” **ICML2024**
> > >
> > > [2] Adrian Bulat et al. “QBB: Quantization with Binary Bases for LLMs.” **NeurIPS2024**
> > >
> > > [3] Elias Frantar and Saleh Ashkboos and Torsten Hoefler and Dan Alistarh. “GPTQ: Accurate Post-Training Quantization for Generative Pre-trained Transformers.” **ICLR2023**
> > >
> > > [4] Sun, Mingjie et al. “Wanda: A Simple and Effective Pruning Approach for Large Language Models.” **ICLR2024**
> > >
> > > [5] Elias Frantar and Dan Alistarh. “SparseGPT: Massive Language Models Can Be Accurately Pruned in One-Shot.” **ICML2023**
> > >
> > > [6] Xin Men et al. "ShortGPT: Layers in Large Language Models are More Redundant Than You Expect." **Arxiv2024**
> >
> >
> > **Q3**: About efficiency and accuracy.
> >
> > > In addition, which techniques contribute the most to efficiency and which method contributes the most to the accuracy?
> >
> > **_Ans for Q3:_**
> >
> > (1) The most important technique for efficiency is the **adaptive N:M sparsity** approach for structured binarization, which allows for a more flexible layer-wise structured binarization. Most LLMs are **memory-intensive models**, which means it is memory-bound. By employing adaptive N:M sparsity, our STBLLM can **alleviate the memory pressure**.
> >
> > (2) The most important technique for accuracy is the **Trisection search** for non-salient weights, which allows for a more **fine-grained partitioning of non-salient weights into three parts**. As shown in Figure 2, the gain of STBLLM over BiLLM largely comes from this our trisection search. For more experimental results, you can also refer to Table2 in the rebuttal of Q2.
> >
> > **Q4**: About practicality and hardware compatibility.
> >
> > > The benchmark results are based on various N:M configurations. However, NVIDIA GPUs mainly support 2:4. The authors may discuss how practical the proposed method is on NVIDIA GPUs.
> >
> > **_Ans for Q4:_**
> >
> > (1) Firstly, we have implemented a **specialized CUDA kernel for structural binarization**, leveraging NVIDIA’s Ampere GPU Sparse Tensor Cores, achieving **17.85x speedup over ABQ-LLM**. This experiments has shown the feasibility for our algorithm.
> >
> > (2) Secondly, regarding various N:M configurations, there are several methods, notably **Vectorized N:M[1]**, that enable **arbitrary N:M ratios on Sparse Tensor Cores**. By converting any N:M format to 2:4 format, it breaks through the limitation of traditional formats on sparsity rate and allows a higher degree of sparsity. This significantly improves memory utilization and computational efficiency. Experiments with VENOM demonstrate that this technique allows for high sparsity without compromising accuracy in modern transformers. These **CUDA kernels in VENOM[1] provide greater flexibility to our STBLLM algorithm**.
> >
> > > **Reference:**
> > >
> > > [1] Castro, Roberto L. et al. “VENOM: A Vectorized N:M Format for Unleashing the Power of Sparse Tensor Cores.” *SC2023*
> >
> > **Finally**, we hope these responses can address the concerns and appreciate the constructive feedback. If the reviewer finds our response adequate, we would appreciate it if the reviewer considers raising the score. We are glad to discuss further comments and suggestions.

---

> > > ### Comment · Reviewer_jQJp · 2024-11-25
> > >
> > > **Q3**:
> > >
> > > Thanks for the clarification. This is very helpful. I think the trisection search is an interesting idea to quantize LLMs by splitting parameters into different groups. I feel the work is a combination of multiple techniques, which is great, but I recommend that the authors highlight which parts are innovated in this work and which ones are adapted.

---

> > > > ### Author Response · Authors · 2024-11-26
> > > > **Response to the Official Comment by Reviewer jQJp(Q3)**
> > > >
> > > > > Q3: For trisection search, I recommend that the authors highlight which parts are innovated in this work and which ones are adapted.
> > > >
> > > > Thanks for your reply. As shown in Figure3(d), the key difference lies in the non-salient weight partition into three partitions. During experiments, we found that increasing the number of paritions can greatly inprove the performance of structured binarization. However, we can not naively apply it to our senario. Here is the pesudo code of the difference between original binary search and our trisection search:
> > > >
> > > > ```
> > > > # BiLLM's implementation: Two Parts
> > > > ## Complexity: O(N)
> > > >
> > > > // BiLLM's implementation: Two Parts
> > > > // Complexity: O(N)
> > > > running_error ← ∞
> > > > best_p ← 0
> > > > for i from 0.1 to 0.9 step (0.8/160) do
> > > >     p1 ← i × max(|W|)
> > > >     (B1, B2) ← Split_by_Alpha(p1)
> > > >     error ← ||W - (B1 + B2)||^2
> > > >     if error < running_error then
> > > >         running_error ← error
> > > >         best_alpha ← alpha_1
> > > >     end if
> > > > end for
> > > >
> > > > // NAIVE implementation Three Parts
> > > > // Complexity: O(N²)
> > > > running_error ← ∞
> > > > best_alpha_1 ← 0
> > > > best_alpha_2 ← 0
> > > > for i from 0.1 to 0.9 step (0.8/160) do
> > > >     for j from 0.1 to 0.9 step (0.8/160) do
> > > >         p1 ← i × max(|W|)
> > > >         p2 ← j × max(|W|)
> > > >         (B1, B2, B3) ← Split_by_Alpha(p1, p2)
> > > >         error ← ||W - (B1 + B2 + B3)||^2
> > > >         if error < running_error then
> > > >             running_error ← error
> > > >             best_p1 ← p1
> > > >             best_p2 ← p2
> > > >         end if
> > > >     end for
> > > > end for
> > > >
> > > > // STBLLM's implementation Three Parts
> > > > // Complexity: O(N)
> > > > running_error ← ∞
> > > > best_p1 ← 0
> > > > best_p2 ← 0
> > > > for i from 0.1 to 0.9 step (0.8/160) do
> > > >     p1 ← i × max(|W|)
> > > >     p2 ← alpha × p1  // Fixed ratio between p1 and p2
> > > >
> > > >     B1 ← Binary(W[|W| > p2])
> > > >     B2 ← Binary(W[p1 < |W| ≤ p2])
> > > >     B3 ← Binary(W[|W| ≤ p1])
> > > >
> > > >     error ← ||W - (B1 + B2 + B3)||^2
> > > >     if error < running_error then
> > > >         running_error ← error
> > > >         best_p1 ← p1
> > > >         best_p2 ← p2
> > > >     end if
> > > > end for
> > > > ```
> > > >
> > > > For the naive implementation, it takes over 6 hours on just LLaMA-2-7B, making it impractial for 70B LLMs. To handle it, we employ a fixed ratio alpha to alleviate the computational burden. Here is the details:
> > > > As shown in Figure 3(c), non-salient weights follow a Gaussian distribution $w \sim \mathcal{N}(\mu, \sigma^2)$, with probability density function:
> > > >
> > > > $f(w) = \frac{1}{\sqrt{2\pi}\sigma} \exp\left(-\frac{w^2}{2\sigma^2}\right)$
> > > >
> > > > Due to the **symmetry and properties of the Gaussian distribution**, we can express the probabilities for each partition:
> > > > (1) Sparse partition: $P_{\text{Sparse}} = 2 \int_{p_2}^\infty f(w) \, dw = 2 \cdot Q\left(\frac{p_2}{\sigma}\right)$
> > > >
> > > > Intermediate partition: $P_{\text{Intermediate}} = 2 \int_{p_1}^{p_2} f(w) \, dw = 2 \left[ Q\left(\frac{p_1}{\sigma}\right) - Q\left(\frac{p_2}{\sigma}\right) \right]$
> > > >
> > > > Dense partition: $P_{\text{Dense}} = \int_{-p_1}^{p_1} f(w) \, dw = 1 - 2 \cdot Q\left(\frac{p_1}{\sigma}\right)$
> > > >
> > > > where $Q(x) = \int_x^\infty \frac{1}{\sqrt{2\pi}} e^{-t^2/2} \, dt$ represents the Gaussian tail probability function. Since our goal is to achieve equal partition areas, we have: $P_{\text{Sparse}} = P_{\text{Dense}} = P_{\text{Intermediate}} = \frac{1}{3}$
> > > > This leads to: $Q\left(\frac{p_1}{\sigma}\right) = \frac{1}{3}$, $Q\left(\frac{p_2}{\sigma}\right) = \frac{1}{6}$, $Q\left(\frac{p_1}{\sigma}\right) - Q\left(\frac{p_2}{\sigma}\right) = \frac{1}{6}$
> > > > Solving these equations:
> > > > $\frac{p_2}{\sigma} = Q^{-1}\left(\frac{1}{6}\right)$, $\frac{p_1}{\sigma} = Q^{-1}\left(\frac{1}{3}\right)$
> > > >
> > > > Using the inverse Q-function values for the standard normal distribution, we can **conclude that $p_2 \approx 2 \times p_1$**, which implies that the **alpha parameter in the above pseudo code equals $2$**.
> > > >
> > > > I hope the above content can address your concerns.

---

> > > > ### Author Response · Authors · 2024-11-26
> > > > **Follow-Up: Updates to Our Rebuttal Based on Your Feedback**
> > > >
> > > > Dear Reviewer jQJp,
> > > >
> > > > Thank you very much for your response to our rebuttal. As requested, we have added two tables to elaborate on the differences between our paper and previous works. Additionally, we have provided detailed explanations regarding the trisection search. If you have any questions, please do not hesitate to reach out.
> > > >
> > > > Best regards,
> > > >
> > > > Authors of Paper #66

---

> ### Author Response · Authors · 2024-11-24
> **Follow-up on Rebuttal to Reviewer jQJp**
>
> Dear Respected Reviewer jQJp,
>
> We sincerely appreciate your thoughtful comments and suggestions, which have greatly helped us improve our work. In response, we have conducted extensive experiments and analyses to thoroughly address all the points you raised. We hope our detailed revisions effectively resolve your concerns. Specifically:
>
> - We have conducted extended experiments over a larger ratio to provide a more comprehensive evaluation of our method.
> - We have clarified our contributions, emphasizing that the residual approximation and trisection search are not part of our primary contributions but are supportive techniques.
> - We have added an additional ablation study to thoroughly analyze the effects of the residual approximation and trisection search, providing more insights into their roles in the overall framework.
> - We have addressed your question regarding how our proposed method can be effectively applied to NVIDIA GPUs, ensuring its practical applicability.
>
> Given that we have not received additional questions or feedback from you over the past few days, we are hopeful that our clarifications have been satisfactory. However, if there are any remaining uncertainties or aspects requiring further discussion, we would be most grateful for the opportunity to address them.
>
> Alternatively, if our revisions have addressed your concerns, we kindly request you to consider updating your evaluation to better reflect the contributions and impact of our work.
>
> Thank you once again for your time and thoughtful consideration.
>
> Best regards,
> Authors of Paper #66

---

> ### Comment · Reviewer_jQJp · 2024-11-25
>
> **Q1**
>
> Thank you for providing the table showing accuracies under various flipping ratios of non-salient parameters. It is very intriguing that BoolQ results are even worse than random guessing (50%). Additionally, flipping non-salient parameters sometimes leads to even better performance than the baselines. I was wondering why we would still want to quantize these parameters instead of pruning them altogether.
>
>
> **Q2**
> I appreciate the authors' responses. Regarding the three key components, I respectively do not see the novelty in the first two. For example, the standardized importance score implies the idea of "importance of each weight by the product of its magnitude and the corresponding input feature norm," which was used on CNNs back in 2016 (e.g., [1]). Furthermore, there are more advanced ways to determine weight importance (e.g., [2,3]). In addition, the layer-wise quantization was also widely used in many quantization and pruning methods. Could the authors elaborate more their innovations?
>
> [1] Hu, H., 2016. Network trimming: A data-driven neuron pruning approach towards efficient deep architectures. arXiv preprint arXiv:1607.03250.
>
> [2] Dong, X., Chen, S. and Pan, S., 2017. Learning to prune deep neural networks via layer-wise optimal brain surgeon. Advances in neural information processing systems, 30.
>
> [3] Frantar, E. and Alistarh, D., 2022. Optimal brain compression: A framework for accurate post-training quantization and pruning. Advances in Neural Information Processing Systems, 35, pp.4475-4488.

---

> > ### Author Response · Authors · 2024-11-26
> > **Response to the Official Comment by Reviewer jQJp (Q1&Q2) (Part1/2)**
> >
> > Dear reviewer jQJp,
> >
> > Thank you for your valuable feedback. We appreciate the opportunity to address your questions in detail:
> >
> > > Q1: Why we would still want to quantize these parameters instead of pruning them together?
> >
> > In fact, the motivation experiments in Figure 1 and Appendix B are conducted under settings with binarized weights. The random weight flipping analysis only applies to binarized weights. Simply pruning the parameters, as done in LLM pruning works like SparseGPT[4] and Wanda[5], is not within the scope of our paper. Regarding the BoolQ performance, the observed degradation under higher random flipping ratios aligns with our expectations - randomly flipping a higher ratio of binarized weights leads to performance degradation.
> >
> > > Q2.1: The standardized importance score implies the idea of "importance of each weight by the product of its magnitude and the corresponding input feature norm," which was used on CNNs back in 2016[1]. Furthermore, there are more advanced ways to determine weight importance (e.g., [2,3]).
> >
> > Thank you for your provided literature, and we have read the work Network Trimming, whose key equation is as follows:
> >
> > To make it clearer, we investigate the difference between these two methods:
> >
> > | Aspect               | STBLLM (SI)                                                                                        | Network Trimming (APoZ)[1]                                                    |
> > | -------------------- | -------------------------------------------------------------------------------------------------- | ----------------------------------------------------------------------------- |
> > | Equation             | $S_{i,j} = \sigma\left(\mu\left(\lvert W_{i,j} \rvert\right)\right) \cdot \lVert X_{:,j} \rVert_2$ | $APoZ_c^{(i)} = \frac{\sum_{k} \sum_{j} f(O_{c,j}^{(i)}(k) = 0)}{N \times M}$ |
> > | Model Applicability  | Applicable to Large Language Models with GeLU activation                                           | Applicable to CNNs with ReLU activation                                       |
> > | Architecture Support | Transformer-based architectures only                                                               | Convolution-based architectures only                                          |
> > | Key Concept          | Mitigates extreme values in weights                                                                | Calculates non-zero fraction of CNN activations                               |
> > | Data Dependency      | Data-free approach (relies only on weights)                                                        | Data-driven approach (requires validation data)                               |
> >
> > We will add this work[1] to background in the revision. For reference[3] - Optimal Brain Compression (OBC), it is a foundamental work in post-training pruning and we have utilized it in Algorithm1, denoted by OBC. As we mentioned in Section 3.2, SparseGPT[4] and GPTQ[6] employ Hessian metric the measure the importance of weights, which is a computational expensive operation. In contrast, we follow a more computational-friendly way to esitmate the importance following Wanda[5]. Regarding reference[2], it is a layer-wise pruning methods, we will elaborate it in Q2.2.

---

> > > ### Author Response · Authors · 2024-11-26
> > > **Response to the Official Comment by Reviewer jQJp (Q1&Q2) (Part2/2)**
> > >
> > > > Q2.2: In addition, the layer-wise quantization was also widely used in many quantization and pruning methods. Could the authors elaborate more their innovations?
> > >
> > > We have to clarify that the second innovation is not about layer-wise quantization, it is about layer-wise semi-structured pruning. To make it more clearer, we list their characters for comparison.
> > >
> > > | Aspect               | STBLLM (Adaptive Layer-wise Binarization) | L-OBS (Layer-wise Optimal Brain Surgeon)                      |
> > > | -------------------- | ----------------------------------------- | ------------------------------------------------------------- |
> > > | Model Type           | Binarized LLM                             | Deep Neural Network                                           |
> > > | Pruning Type         | N:M structured pruning                    | Unstructured pruning                                          |
> > > | Assignment           | Assign from N:M (N<M)                     | Assign freely due to the feature of unstructured pruning      |
> > > | Flexibility          | Less flexible                             | More flexible                                                 |
> > > | Search Strategy      | DominoSearch                              | Using Hessian and backpropagation to control layer-wise error |
> > > | Training Requirement | Without retraining                        | Require iterative retraining                                  |
> > >
> > > [1] Hu, H., 2016. Network trimming: A data-driven neuron pruning approach towards efficient deep architectures. arXiv preprint arXiv:1607.03250.
> > >
> > > [2] Dong, X., Chen, S. and Pan, S., 2017. Learning to prune deep neural networks via layer-wise optimal brain surgeon. Advances in neural information processing systems, 30.
> > >
> > > [3] Frantar, E. and Alistarh, D., 2022. Optimal brain compression: A framework for accurate post-training quantization and pruning. Advances in Neural Information Processing Systems, 35, pp.4475-4488.
> > >
> > > [4] Frantar, Elias and Dan Alistarh. "SparseGPT: Massive Language Models Can Be Accurately Pruned in One-Shot." ICML2023
> > >
> > > [5] Sun, Mingjie et al. "A Simple and Effective Pruning Approach for Large Language Models." arXiv preprint arXiv:2306.11695
> > >
> > > [6] Frantar, Elias et al. "GPTQ: Accurate Post-training Compression for Generative Pretrained Transformers." arXiv preprint arXiv:2210.17323. ICLR23
> > >
> > > I hope the above answers can make it clearer.

---

> ### Author Response · Authors · 2024-11-28
> **Request for Review Feedback**
>
> Dear Reviewer jQJp,
>
> I hope this message finds you well.
>
> I am writing to kindly follow up regarding our manuscript (#66) that is currently under review. We truly understand that you have many commitments, and we greatly appreciate the time and effort you’ve already dedicated to evaluating our work.
>
> As it has been over eight days since the submission, and with the deadline approaching in about four days, we would be incredibly grateful if you could find a moment to provide your feedback. Your insights are invaluable to us, and we deeply appreciate your contribution to the review process.
>
> Thank you so much for your time and consideration. We fully understand how busy you must be and truly appreciate any attention you can give to our manuscript.
>
> With sincere thanks,
>
> Authors of #66

---

> ### Author Response · Authors · 2024-12-01
> **Last Day Reminder to Reviewer jQJp**
>
> Dear Reviewer jQJp,
>
> Tomorrow is the final day for discussion regarding our paper. Over the past few weeks, we have not received any feedback from you, and we are unsure of the reason. As a reviewer, it is essential to provide feedback, even if it may be critical. We genuinely value your insights and believe that your comments will contribute greatly to improving the quality of our work.
>
> We kindly request that you take a moment to share your thoughts with us before the deadline. Your efforts are instrumental in advancing the review process for ICLR.
>
> Thank you for your time and consideration.
>
> Best regards,
>
> Authors of #66

---

### Author Response · Authors · 2024-11-19
**Thanks to all Reviewers for Recognition and Constructive Comments**

Dear Reviewers, Area Chairs, and Program Chairs,

We sincerely thank all reviewers for their positive feedback and constructive comments. The reviewers have acknowledged the novelty, impact, superior performance, and efficient implementation of our method. Below, we summarize the strengths of our paper as highlighted by the reviewers:

**[Novelty & Methodology]:**

- **Reviewer jQJp**: The work presents a structural binarization method for LLMs by combining N:M sparsity, residual approximation, and block-wise error compensation. The analysis on flipping non-salient binarized weights is intriguing.
- **Reviewer KgJf**: The paper proposes a structured binary quantization method to accelerate LLM inference, combining N:M pruning and binary quantization, and introducing a new SI method for identifying significant weights.
- **Reviewer 1cy6**: The framework combines pruning and binarization to compress large, post-trained models, introducing a new Standardized Importance (SI) metric that avoids costly second-order computations.
- **Reviewer yy1T**: A new metric matrix is proposed to represent the importance of different weights, allowing for effective sparsification and quantization.

**[Impactful & Performance]:**

- **Reviewer jQJp**: The proposed method achieves the lowest perplexity among all compared methods in the sub-1-bit regime.
- **Reviewer KgJf**: Experimental results demonstrate that STBLLM outperforms BiLLM under the same bit budget, achieving significant performance improvement.
- **Reviewer 1cy6**: The method significantly reduces computational costs, accelerates inference, and maintains strong performance.
- **Reviewer yy1T**: The method achieves superior performance at higher compression ratios, integrating sparsity with quantization for efficient inference.

**[Efficient Implementation]:**

- **Reviewer jQJp**: A specialized CUDA kernel is designed to support structural binarization, achieving a 17.85x speedup over ABQ-LLM's 2-bit implementation.
- **Reviewer KgJf**: Dedicated CUDA implementations for the proposed method result in significant performance improvement.
- **Reviewer yy1T**: A dedicated CUDA kernel was developed to optimize the performance of the sparse and quantized model on GPU hardware, enabling efficient memory access patterns and computation.

**[Good Presentation]:**

- **Reviewer 1cy6**: The paper is well-organized and easy to follow, with a clearly stated problem. The approach is logical and rigorous, fully validating its effectiveness through comprehensive experiments.

In the past few days, we have conducted **additional experiments, clarified points, and engaged in discussions** to address the valuable comments provided by the reviewers. Based on the constructive feedback, we have **carefully revised the manuscript of our work using yellow highlights**. We hope our detailed responses can alleviate the concerns.

Best regards and thanks,

Authors of #66

---

> ### Comment · Reviewer_yy1T · 2024-11-19
> **RE: The Summary of Our Responses to All Official Reviews**
>
> From the perspective of an ICLR reader rather than an official reviewer, I find myself curious about the decision to focus solely on emphasizing the strengths of the paper in the section titled “Summary of Our Responses to All Official Reviews.” Are there truly no potential weaknesses that warrant mention in the summary? Considering that all reviewers’ feedback is publicly available, I wonder if re-summarizing the strengths serves to draw reviewers’ attention, or if it might inadvertently come across as misleading.
>
> **Please note that I have no intention to offend the authors. My comments are merely an attempt to better understand the rationale behind this strategy and to ensure the summary is as balanced and transparent as possible.**

---

> ### Author Response · Authors · 2024-11-19
> **Thanks for the Suggestion and Look Forward to Further Feedback**
>
> Dear Reviewer yy1T,
>
> Thank you for your insightful suggestions and feedback on our paper. We greatly appreciate the time and effort you have dedicated to reviewing our work.
>
> Over the past few days, we have made our best effort to address each reviewer's concerns individually, providing clarifications and conducting additional experiments. As we do not provide individual detailed responses to each reviewer's recognitions, we have collated and organized these comments in this section. As per your suggestion, we have revised the title of this section to avoid any potential confusion for the readers.
>
> We look forward to you reviewing our response, and we will actively address any further questions or concerns you may have. Once again, thank you for your valuable input, which has helped us improve our work.
>
> Best regards,
>
> Authors of #66

---

### Comment · Area_Chair_Bhn3 · 2024-11-25
**Please engage in discussions**

Dear all,

Many thanks to the reviewers for their constructive reviews and the authors for their detailed responses.
Please use the next ~2 days to discuss any remaining queries as the discussion period is about to close.

thank you.

Regards,

AC

---

### Meta-Review · Area_Chair_Bhn3 · 2024-12-17

**Metareview:**

The paper introduces STBLLM, a new method for compressing large language models (LLMs) to less than 1-bit precision, i.e. the paper's main claim. The main strength of the paper lies in its structured binarisation method, which employs N:M sparsity and fine-grained weight quantisation to achieve sub-1-bit compression. This method allows for more efficient storage and computation compared to previous techniques like BiLLM and PB-LLM. One of the main contributions is the Standardised Importance (SI) metric, which estimates weight importance without the need for computationally expensive Hessian-based methods. By considering weight magnitude and input feature norm, the SI metric allows for more effective weight pruning and sparsification.

Another strength is the layer-wise adaptive binarisation, which enables different layers of the LLM to have varying N:M sparsity ratios. This approach achieves a better balance between compression and model accuracy. Weights are further divided into sparse, intermediate, and dense regions, with each region undergoing a unique quantisation scheme. This fine-grained approach ensures that critical weights are preserved while less important weights are aggressively compressed.

The paper also demonstrates practical efficiency improvements through the use of a specialised CUDA kernel for structured binarisation, which is optimised for NVIDIA's Ampere sparse tensor cores. This optimisation results in a 17.85x speedup compared to existing 2-bit implementations. Some empirical validation on LLaMA (1, 2, and 3), OPT, and Mistral models confirms the method's claims. Across zero-shot, perplexity, and efficiency metrics, STBLLM consistently outperforms other methods like BiLLM and PB-LLM, achieving superior compression with minimal impact on model performance.

Despite its strengths, the paper has several limitations, as highlighted by the reviewers. One of them is that STBLLM does not support Mixture of Experts (MoE) models or Mamba-based LLMs, which limits its scope; however, it is to be expected in my opinion. Additionally, the approach introduces considerable complexity due to its reliance on multiple stages of partitioning, trisection, and residual binarisation.

Another potential limitation is that while the method demonstrates improved performance on perplexity and zero-shot evaluations, the impact on real-world downstream tasks (like question answering or multi-step reasoning) is not discussed in detail. This raises questions about the generalisation of STBLLM in practical applications.

Finally, while the authors highlight the robustness of the approach to random weight flipping, the paper shows that performance fluctuations occur as the flipping ratio increases. While the approach appears stable in controlled experiments, it is unclear how robust it would be in real-world scenarios where noise or hardware faults might cause weight flips. Furthermore, while STBLLM claims to “break the 1-bit barrier,” the paper does not adequately stress the theoretical limits of this approach.

Having said all the above, I do believe the paper has been improved since its original submission. I would encourage the authors to add any remaining revisions to the paper based on the rebuttal for the camera ready version.

**Additional Comments On Reviewer Discussion:**

The reviewers scored this with 5, 5, 6 and 8. A couple of reviewers did not engage with the rebuttal process, and one of them yy1T only responded to my thread. KgJf raised some concerns regarding the paper's novelty which were thoroughly addressed by the authors.
I also find yy1T's approach to not respond to authors' rebuttal but instead only respond to my thread and authors' complementary summary a bit unconventional. I think the authors have gone above and beyond addressing the reviewers' comments within reason.

---

### Decision · Program_Chairs · 2025-01-22

Accept (Poster)